# EARTHQUAKENPP: BENCHMARK DATASETS FOR EARTHQUAKE FORECASTING WITH NEURAL POINT PROCESSES

## ABSTRACT

Classical point process models, such as the epidemic-type aftershock sequence (ETAS) model, have been widely used for forecasting the event times and locations of earthquakes for decades. Recent advances have led to Neural Point Processes (NPPs), which promise greater flexibility and improvements over classical models. However, the currently-used benchmark dataset for NPPs does not represent an up-to-date challenge in the seismological community since it lacks a key earthquake sequence from the region and improperly splits training and testing data. Furthermore, initial earthquake forecast benchmarking lacks a comparison to state-of-the-art earthquake forecasting models typically used by the seismological community. To address these gaps, we introduce EarthquakeNPP: a collection of benchmark datasets to facilitate testing of NPPs on earthquake data, accompanied by a credible implementation of the ETAS model. The datasets cover a range of small to large target regions within California, dating from 1971 to 2021, and include different methodologies for dataset generation. In a benchmarking experiment, we compare three spatio-temporal NPPs against ETAS and find that none outperform ETAS in either spatial or temporal log-likelihood. These results indicate that current NPP implementations are not yet suitable for practical earthquake forecasting. EarthquakeNPP also provides generative evaluation metrics, enabling broader model classes to be benchmarked and facilitating the future collaboration between the seismology and machine learning communities.

## 1 INTRODUCTION

Operational earthquake forecasting by global governmental organisations such as the US Geological Survey (USGS) necessitates the development of models which can forecast the times and locations of damaging earthquakes. While model development is ongoing in the seismology community, recent improvements have relied upon refinement of a spatio-temporal point process model known as the Epidemic-Type Aftershock Sequence (ETAS) model (Ogata, 1988; 1998), despite significant growth in available data (Takanami et al., 2003; Shelly, 2017; Ross et al., 2019; White et al., 2019; Mousavi et al., 2020; Tan et al., 2021; Mousavi & Beroza, 2023).

In contrast, the machine learning community has offered promising advancements over classical point process models like ETAS with Neural Point Process (NPP) models, showcasing greater flexibility (Du et al., 2016; Omi et al., 2019a; Shchur et al., 2019; Jia & Benson, 2019; Chen et al., 2021; Zhou et al., 2022; Zhou & Yu, 2024). While some initial benchmarking of these models has been conducted on an earthquake dataset in Japan, these experiments lack relevance for stakeholders in the seismology community. The benchmark lacks a key earthquake sequence from the region, fails to recreate an operational setting with proper train-test splits, and doesn't compare against state-of-the-art models like ETAS.

Here, we introduce EarthquakeNPP: a curated collection of datasets designed for benchmarking NPP models in earthquake forecasting, accompanied by a state-of-the-art benchmark model. These datasets are derived from publicly available raw data, which we process and configure within our platform to facilitate meaningful forecasting experiments relevant to stakeholders in the seismology community. Covering various regions of California, these datasets represent typical forecasting zones

Table 1: Comparison of EarthquakeNPP datasets with the existing NPP benchmark dataset for earthquakes.

| Dataset | Chronological Training/Test Splits | Complete Timespan | Complete Magnitudes | Used by Local Agencies |
|---|---|---|---|---|
| Chen et al. (2021) Dataset | ✗ | ✗ | ✗ | ✗ |
| EarthquakeNPP Datasets | ✓ | ✓ | ✓ | ✓ |

and encompass data commonly utilized by forecast issuers. Moreover, employing modern techniques, some datasets include smaller magnitude earthquakes, exploring the potential of numerous small events to enhance forecasting performance through flexible NPPs. To unify efforts, we present an operational-level implementation of the ETAS model alongside the datasets, serving as a benchmark for NPPs.

Although initial benchmarking finds that none of the 3 tested NPP implementations outperform ETAS, EarthquakeNPP aims to serve as a platform for future NPP development. The platform facilitates the generative evaluation procedure used for rigorous benchmarking in the seismology community. This directs the impact of future NPPs to stakeholders in seismology and broadens the scope of models beyond NPPs (e.g. times series models (Wang et al., 2017), Bayesian approaches (Serafini et al., 2023)). Access to the dataset collection, along with comprehensive documentation and notebooks, can be found at `https://anonymous.4open.science/r/EarthquakeNPP-2D51`.

## 1.1 RELATED WORK

**Benchmarking by the NPP Community.** Chen et al. (2021) introduced an earthquake dataset for benchmarking the Neural Spatio-temporal Point Process (NSTPP) model using a global dataset from the U.S. Geological Survey, focusing on Japan from 1990 to 2020. They considered earthquakes with magnitudes above 2.5, splitting the data into month-long segments with a 7-day offset. They exclude earthquakes from November 2010 to December 2011, deeming these sequences "too long" and "outliers." However, this period includes the 2011 Tohoku earthquake (Mori et al., 2011), the largest earthquake recorded in Japan and the fourth largest in the world, at magnitude 9.0. This exclusion renders the benchmarking experiment irrelevant for seismologists, as it is precisely these large earthquakes and their aftershocks that are crucial to forecast due to their damaging impact. Additionally, these events are of significant scientific interest because they provide valuable insights into the earthquake rupture process.

The dataset segments are divided for training, testing, and validation. Instead of a chronological partitioning that mirrors operational forecasting, the segments are assigned in an alternating pattern. This approach misrepresents a realistic forecasting scenario and inflates performance measures due to earthquake triggering (Freed, 2005). Since the model is tested on windows immediately preceding training windows, it exploits causal dependencies backwards it time.

Although earthquakes with magnitudes above 2.5 are considered by Chen et al. (2021), following a change in USGS policy on global data collection, from 2009 onwards, only events above magnitude 4.0 are recorded in the dataset. For earthquake forecasting in Japan, seismologists use datasets from Japanese data centers since they are more comprehensive and complete than global datasets. Section A.2 describes the biases incurred from such data missingness.

Chen et al. (2021) benchmark their model against another spatio-temporal model, Neural Jump SDEs (Jia & Benson, 2019), and a temporal-only Hawkes process, even though a spatio-temporal Hawkes process would provide a more rigorous benchmark. Subsequent papers adopting this benchmark (Zhou et al., 2022; Yuan et al., 2023; Zhou & Yu, 2024) similarly lack comparisons to a spatio-temporal Hawkes process, benchmarking instead against temporal-only or spatial-only baselines or other spatio-temporal NPPs.

**Benchmarking by the Seismology Community.** Model comparison has been crucial in the development of earthquake forecasting models since their inception (Kagan & Knopoff, 1987; Ogata, 1988). The Collaboratory for the Study of Earthquake Predictability (CSEP) (Michael & Werner, 2018; Schorlemmer et al., 2018; Savran et al., 2022; Iturrieta et al., 2024) (`https://cseptesting.org/`) aims to unify the framework for earthquake model testing and evaluation, hosting retrospective

and fully prospective forecasting experiments globally. CSEP benchmarks short-term models using performance metrics that require forecasts to be generated by simulating many repeat sequences over a specified time horizon (typically one day). These simulated forecasts are compared by discretizing time and space intervals, with test statistics calculated for event counts, magnitudes, locations, and times. The simulation-based approach allows the inclusion of generative models that don't output explicit earthquake probabilities (i.e., a likelihood), and enables evaluation of the full distribution of entire sampled sequences.

Two existing works benchmark NPPs for earthquake forecasting within the seismology community. The first by Dascher-Cousineau et al. (2023) extends a temporal-only NPP from Shchur et al. (2019) to include earthquake magnitudes. The second by Stockman et al. (2023) extends another temporal-only model by Omi et al. (2019a) to target larger magnitude events. Both models are benchmarked against a temporal ETAS model, showing moderate improvements over the baseline. Extending these models to include spatial data is necessary for further testing and potential operational use in the seismological community.

## 1.2 SCOPE OF THIS WORK

Since generating repeated sequences over forecast horizons is computationally costly, the seismology community uses the mean log-likelihood on held-out data for a more streamlined metric during model development (Ogata, 1988; Harte, 2015). Our platform uses this metric in the NPP benchmarking experiment and provides detailed guidance on CSEP's simulation-based procedure, enabling future NPP implementations and evaluations within CSEP experiments.

This work aims to bridge Machine Learning and seismology by establishing a baseline for comparing NPP models to state-of-the-art, domain-based models. Only NPPs capable of generating log-likelihoods are within scope, as no valid score exists for models lacking this capability (e.g. Yuan et al., 2023; Li et al., 2023). Traditional metrics like Root Mean Square Error (RMSE) and Mean Absolute Error (MAE) are inadequate and potentially misleading for seismological predictions (Hodson, 2022), as earthquake occurrence follows power law distributions (Kagan, 1994; Felzer & Brodsky, 2006) that are heavy-tailed, making the errors non-Gaussian and non-Laplacian — contrary to the assumptions underlying RMSE and MAE (see Section G). To ensure seismological relevance, we challenge authors of NPP models to implement forecasts using CSEP's evaluation framework and benchmark their results against the performance of the ETAS model.

## 2 BACKGROUND

### 2.1 SPATIO-TEMPORAL POINT PROCESSES

A spatio-temporal point process is a continuous-time stochastic process that models the random number of events $N(S \times (t_a, t_b])$ which occur in a space-time interval $S \times (t_a, t_b]$, $S \in \mathbb{R}^2$, $(t_a, t_b] \in \mathbb{R}^+$. This process is typically defined by a non-negative *conditional intensity function*

$$\lambda(t, \mathbf{x}|\mathcal{H}_t) := \lim_{\Delta t, \Delta \mathbf{x} \to 0} \frac{\mathbb{E}\left[N([t, t + \Delta t) \times B(\mathbf{x}, \Delta \mathbf{x})|\mathcal{H}_t\right]}{|B(\mathbf{x}, \Delta \mathbf{x})|}, \tag{1}$$

where $\mathcal{H}_t = \{(t_i, \mathbf{x}_i)|t_i < t\}$ denotes the history of events preceding time $t$ and $|B(\mathbf{x}, \Delta \mathbf{x})|$ is the Lebesgue measure of the ball $B(\mathbf{x}, \Delta \mathbf{x})$ with radius $\Delta \mathbf{x}$. Given we observe a history of events up to $t_i$, the probability density function (pdf) of observing an event at time $t$ and location $\mathbf{x}$ is given by

$$p(t, \mathbf{x}|\mathcal{H}_{t_i}) = \lambda(t, \mathbf{x}|\mathcal{H}_{t_i}) \cdot \exp\left(-\int_{t_i}^{t} \int_{S} \lambda(s, \mathbf{z}|\mathcal{H}_s) d\mathbf{z} ds\right). \tag{2}$$

Most models specify the conditional intensity function, though some (e.g. Shchur et al., 2019; Chen et al., 2021; Yuan et al., 2023) directly model this pdf. Model parameters are typically estimated by maximizing the log-likelihood of observed events within a training time interval $[T_0, T_1]$ and spatial region $S$,

$$\log p(\mathcal{H}_T) = \underbrace{\sum_{i=0}^{n} \log \lambda(t_i|\mathcal{H}_{t_i}) - \int_{T_0}^{T_1} \int_{S} \lambda(s, \mathbf{z}|\mathcal{H}_s) d\mathbf{z} ds}_{\text{Temporal log-likelihood}} + \underbrace{\sum_{i=0}^{n} \log f(\mathbf{x}_i|t_i, \mathcal{H}_{t_i})}_{\text{Spatial log-likelihood}}, \tag{3}$$

where the decomposition of the spatio-temporal conditional intensity function, $\lambda(t_i, \mathbf{x}_i | \mathcal{H}_{t_i}) = \lambda(t_i | \mathcal{H}_{t_i}) \cdot f(\mathbf{x}_i | t_i, \mathcal{H}_{t_i})$, allows the log-likelihood to be written as contributions from the temporal and spatial components. In practice, this exact function is often not maximized directly during training: for models specified through the conditional intensity function, an analytical solution to the integral term is generally not possible and is approximated numerically.

For model evaluation and comparison, the log-likelihood of observing events in the test set can be used as a performance metric. This is consistent with a wealth of literature in the seismology community (see Zechar et al., 2010, and references therein) as well as the wider general point process literature (Daley & Vere-Jones, 2004), which now includes neural point processes (Shchur et al., 2021). The metric evaluates models that output probability distributions over their predictions and consequently penalises models that are overconfident. Although evaluating on events in the test set, the test log-likelihood, $\log p\left((t_i, \mathbf{x}_i) | t_i \in [T_2, T_3], \mathcal{H}_{T_2}\right)$, may still contain dependence upon events prior to the test window $[T_2, T_3]$, typically contained in the history $\mathcal{H}_{T_2}$ of the intensity function. Comparing the mean log-likelihood per event provides the *information gain* from one model to another (Daley & Vere-Jones, 2004).

Point processes are the dominant modeling approach in the seismology community, used extensively in both real-time operational earthquake forecasting (Mizrahi et al., 2024a) and established benchmarking experiments (CSEP) (Taroni et al., 2018; Rhoades et al., 2018). The point process representation of earthquake data aligns naturally with their occurrence as discrete events in time (Kagan, 1994). Furthermore, this modeling approach is favored over discretized forecasting models (e.g., time series) because it eliminates the need for optimizing binning strategies and allows for immediate updates, rather than waiting until the end of a time bin — a delay that could miss critical, potentially damaging events.

## 2.2 ETAS

The Epidemic Type Aftershock Sequence (ETAS) model (Ogata, 1998) is a spatio-temporal Hawkes process which models how earthquakes cluster in time and space. It has been adopted for operational earthquake forecasting by government agencies in California (Milner et al., 2020), New-Zealand (Christophersen et al., 2017), Italy (Spassiani et al., 2023), Japan (Omi et al., 2019b) and Switzerland (Mizrahi et al., 2024b), and performs consistently well in CSEP's retrospective and fully prospective forecasting experiments (e.g. Woessner et al., 2011; Rhoades et al., 2018; Taroni et al., 2018; Cattania et al., 2018; Mancini et al., 2019; 2020; 2022). The general formulation of the model is

$$\lambda(t, \mathbf{x} | \mathcal{H}_t; \theta) = \mu + \sum_{i:t_i < t} g(t - t_i, ||\mathbf{x} - \mathbf{x}_i||_2^2, m_i), \tag{4}$$

where $\mu$ is a constant background rate of events, $g(\cdot, \cdot, \cdot)$ is a non-negative excitation kernel which describes how past events contribute to the likelihood of future events and $m_i$ are the associated magnitudes of each event. The equivalent formulation as a Hawkes branching process accompanies a causal branching structure $\mathbf{B}$. This concept broadly aligns with the understanding of the physics of earthquake triggering and interaction, e.g. via dynamic wave triggering (Brodsky & van der Elst, 2014) and static stress triggering (Gomberg, 2018; Mancini et al., 2020).

Although ETAS can be fit by maximizing the log-likelihood function directly, parameter estimation is typically performed by simultaneously estimating the branching structure $\mathbf{B}$. Veen & Schoenberg (2008) developed an Expectation Maximisation (EM) procedure, which maximises the marginal likelihood over the unobserved branching structure, $\log \int p(\mathcal{H}_{T_1} | \mathbf{B}, \theta) p(\mathbf{B} | \theta) d\mathbf{B}$ through the iteration

$$\theta^{(k+1)} = \arg\max_{\theta} \mathbb{E}_{\mathbf{B} \sim p(\cdot | \mathcal{H}_{T_1}, \theta^{(k)})} \left[ \log p(\mathcal{H}_{T_1}, \mathbf{B} | \theta) \right]. \tag{5}$$

This avoids the need to numerically approximate the integral term in the likelihood, provides more stability during estimation and simultaneously distinguishes background events from triggered events.

The formulation of the ETAS model we present with the EarthquakeNPP datasets is implemented in the `etas` python package by Mizrahi et al. (2022). It defines the triggering kernel as

$$g(t, r^2, m) = \frac{e^{-t/\tau} \cdot k \cdot e^{a(m-M_c)}}{(t+c)^{1+\omega} \cdot \left(r^2 + d \cdot e^{\gamma(m-M_c)}\right)^{1+\rho}}, \tag{6}$$

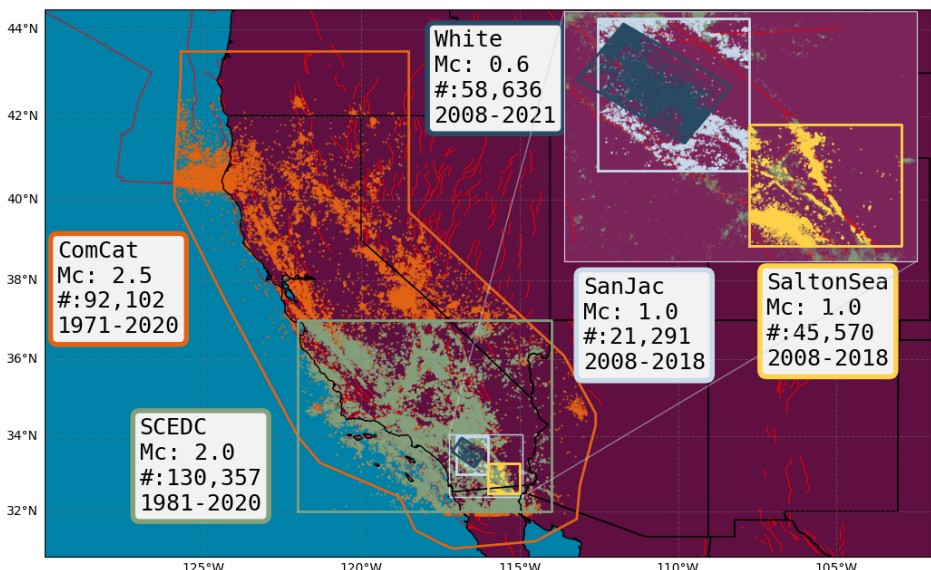

Figure 1: Earthquakes contained in the observational datasets found in EarthquakeNPP. Colours indicate the respective datasets, including the target region, magnitude of completeness $M_c$, number of events and the time period that the dataset spans. In red is a fault map from the GEM Global Active Faults Database (Styron & Pagani, 2020).

where $r^2$ is the squared distance between events and $k, a, c, \omega, \tau, d, \gamma, \rho$ are the learnable parameters along with the constant background rate $\mu$. This triggering kernel is derived from statistical distributions found through decades of observational studies (Utsu & Seki, 1955; Utsu, 1970; Utsu et al., 1995) and several of the learnable parameters have been linked to physical properties of the earthquake rupture process (Utsu et al., 1995; Ide, 2013).

## 3 EARTHQUAKENPP DATASETS

The EarthquakeNPP datasets encompass earthquake records, including timestamps, geographical coordinates, and magnitudes, documented within California from 1971 to 2021. California, with its dense network and high seismic hazard, has been extensively studied, demonstrating the utility of forecasting algorithms (Gerstenberger et al., 2004; Field, 2007; Field et al., 2021). It encompasses the San Andreas fault plate boundary system (Zoback et al., 1987) and includes modern high-resolution catalogs with numerous small magnitude earthquakes, offering potential for new, more expressive models.

Notebooks to access and preprocess these public datasets along with the associated benchmarking experiment are publicly accessible at https://anonymous.4open.science/r/EarthquakeNPP-2D51, accompanied by more detailed documentation for each dataset. A summary of how earthquake datasets are generated, along with the associated challenges of using earthquake catalog data can be found in Appendix A. Table 2 provides a short summary of each EarthquakeNPP dataset.

## 4 BENCHMARKING EXPERIMENT

We now use EarthquakeNPP to benchmark three spatio-temporal NPPs with prior positive claims on earthquake forecasting.

**Neural Spatio-Temporal Point Process (NSTPP)** (Chen et al., 2021): a pdf based NPP that parameterizes the spatial pdf with continuous-time normalizing flows (CNFs). We use their Attentive CNF model for its computational efficiency and overall performance versus their other model Jump CNF (Chen et al., 2021).

Table 2: Summary of EarthquakeNPP datasets, including: region, dataset development, magnitude threshold ($M_c$), number of training (combined with validation) events, and number of testing events. The chronological partitioning of training, validation, and testing periods is also detailed. An auxiliary (burn-in) period begins from the **Start** date, followed by the respective starts of the training, validation, and testing periods. All dates are given as 00:00 UTC on January 1st, unless noted (* refers to 00:00 UTC on January 17th). Finally, we give our purpose for including each dataset.

| | ComCat | SCEDC | White | QTM |
|---|---|---|---|---|
| **Region** | Whole of California | Southern California | San Jacinto Fault-Zone | `QTM_SanJac:` San Jacinto Fault-Zone, `QTM_SaltonSea:` Salton Sea |
| **Development** | The U.S. Geological Survey (USGS) National Earthquake Information Center (NEIC) monitors global earthquakes (Mw 4.5 or larger) and provides complete seismic monitoring of the United States for all significant earthquakes (> Mw 3.0 or felt). Its contributing seismic networks have produced the Advanced National Seismic System (ANSS) Comprehensive Catalog of Earthquake Events and Products. | The Southern California Seismic Network (SCSN) has developed and maintained the standard earthquake catalog for Southern California (Hutton et al., 2010) since the Caltech Seismological Laboratory began routine operations in 1932. Significant network improvements since the 1970s and 1980s reduced the catalog completeness from Mw 3.25 to Mw 1.8. | White et al. (2019) created an enhanced catalog focusing on the San Jacinto fault region, using a dense seismic network in Southern California. This denser network, combined with automated phase picking (STA/LTA), ensures a 99% detection rate for earthquakes greater than Mw 0.6 in a specific subregion (White et al., 2019). | Using data collected by the SCSN, Ross et al. (2019) generated a denser catalog by reanalyzing the same waveform data with a template matching procedure that looks for cross-correlations with the wavetrains of previously detected events. |
| **$M_c$** | Mw 2.5 | `SCEDC_20:` Mw 2.0, `SCEDC_25:` Mw 2.5, `SCEDC_30:` Mw 3.0 | Mw 0.6 | Mw 1.0 |
| **# Train/Test Events** | 79,037 / 23,059 | `SCEDC_20:` 128,265 / 14,351, `SCEDC_25:` 43,221 / 5,466, `SCEDC_30:` 12,426 / 2,065 | 38,556 / 26,914 | `QTM_SanJac:` 18,664 / 4,837, `QTM_SanJac:` 44,042 / 4,393 |
| **Start-Train-Val-Test-End** | 1971-1981-1998-2007-2020* | 1981-1985-2005-2014-2020 | 2008-2009-2014-2016-2018 | 2008-2009-2014-2016-2018 |
| **Purpose** | Example of data currently in use for operational forecasting (USGS utilizes ComCat in aftershock forecasts they release to the public.) | Three magnitude thresholds (Mw 2.0, 2.5, 3.0) explore the effect of truncation on forecasting model performance. | To explore if newly detected low magnitude earthquakes contain additional predictive information. | To explore if newly detected low magnitude earthquakes contain additional predictive information (with different detection methodology). |

**Deep Spatio-Temporal Point Process (Deep-STPP)** (Zhou et al., 2022)**:** a conditional intensity function based NPP that constructs a non parametric space-time intensity function governed by a deep latent process. The intensity function enjoys a closed form integration, avoiding the need for numerical approximation.

**Automatic Integration for Spatiotemporal Neural Point Processes (AutoSTPP)** (Zhou & Yu, 2024)**:** a conditional intensity function based NPP which jointly models the 3D space-time integral of the intensity along with its derivative (the intensity function) using a dual network approach.

The benchmark is against the **ETAS** model defined in section 2.2, as well as a homogeneous **Poisson** process. The Poisson model is fit to events in the auxiliary, training and validation windows to provide a baseline score against which to compare all four other models.

Validation is typically not part of the estimation procedure for ETAS, so it is fit using the combined training and validation windows. NPPs follow the standard training/validation/testing procedure of machine learning. When possible, a model's likelihood for training, validation, and testing can depend on events occurring before the splits through memory in its history. The exception is NSTPP, lacking

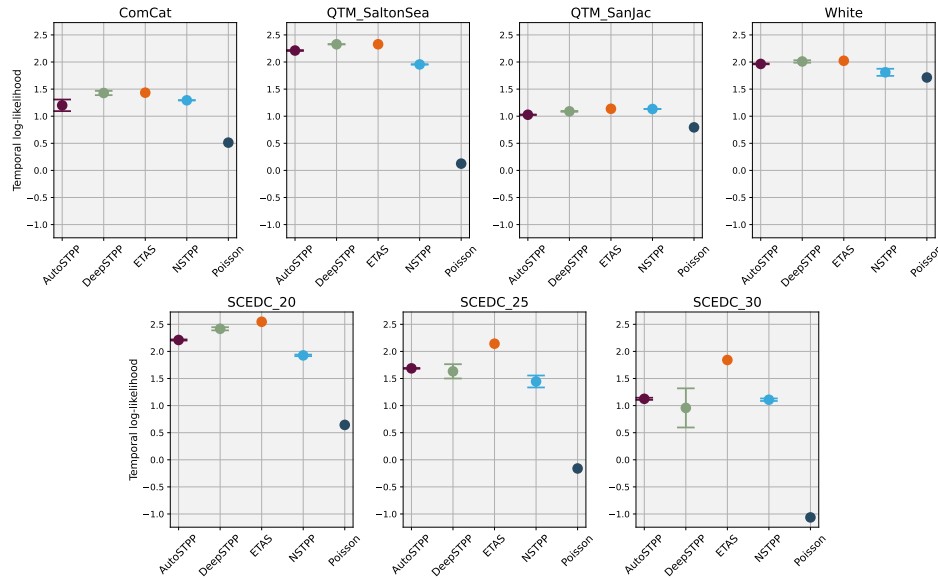

Figure 2: Test temporal log-likelihood scores for all the spatio-temporal point process models on each of the EarthquakeNPP datasets. Error bars of the mean and standard deviation are constructed for the NPPs using three repeat runs.

a direct dependency on prior events. Nonetheless, its likelihood is evaluated on the same events as the other models. The definition of the ETAS model (equation 4) specifies how the magnitudes of earthquakes in the history contribute towards the intensity function. This earthquake magnitude dependence is not implemented in any of the NPPs we benchmark, since it requires modeling choices beyond the scope of this work.

Figures 2 and 3 present the temporal and spatial log-likelihood scores of all models on the EarthquakeNPP datasets. The ETAS model consistently achieves the highest temporal and spatial log-likelihood across all datasets, though some NPP models demonstrate comparable temporal performance on the `ComCat`, `QTM_SaltonSea`, `QTM_SanJac`, and `White` catalogs. Among the NPP models, Deep-STPP generally exhibits the best temporal log-likelihood, likely due to its formulation, which accounts for the influence of unobserved events—a phenomenon that varies temporally in earthquake data (see Section A.2). In contrast, AutoSTPP achieves the highest spatial log-likelihood, attributed to its ability to capture anisotropic Hawkes kernels (see Figure 2 of Zhou & Yu (2024)), which are often observed in earthquake data (Page & van der Elst, 2022).

The improved relative temporal performance of all NPPs compared to ETAS, particularly when the magnitude threshold is lowered from 3.0 to 2.0 in the `SCEDC` dataset, indicates that low magnitude earthquakes provide valuable predictive information for NPPs. This is further suggested by the comparable performance of NPPs to ETAS on low-magnitude catalogs such as `QTM_SaltonSea`, `QTM_SanJac`, and `White`. The stronger temporal performance of NPPs on datasets such as `ComCat`, `QTM_SaltonSea`, `QTM_SanJac`, and `White` may also reflect their ability to model more complex physical processes, such as earthquake swarms (Llenos & van der Elst, 2019) or tectonic activity near the Mendocino Triple Junction (Hellweg et al., 2024). Additional datasets and results are included in Appendix B.

## 5 CSEP CONSISTENCY TESTS

EarthquakeNPP also supports the earthquake forecast evaluation protocol developed by the Collaboratory for the Study of Earthquake Predictability (CSEP). In this procedure a model generates 24-hour forecasts through 10,000 repeat simulations of earthquake sequences at the beginning of every day in the testing period. This procedure mimics how earthquake forecasts are generated in an operational

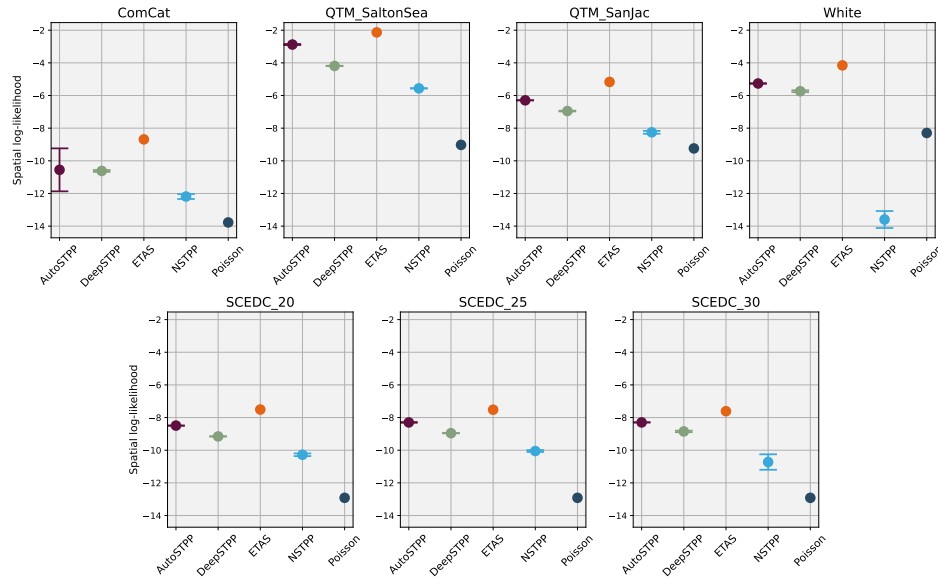

Figure 3: Test spatial log-likelihood scores for all the spatio-temporal point process models on each of the EarthquakeNPP datasets. Error bars of the mean and standard deviation are constructed for the NPPs using three repeat runs.

setting (van der Elst et al., 2022). Models can then be evaluated by comparing the observed sequence with the distribution over model simulations. Three test statistics target the temporal, spatial and magnitude components of the forecasts, where a test is failed if the observed statistic falls within a pre-defined rejection region (Figure 4). We demonstrate this procedure for the ETAS model and report performance scores as a benchmark for future implementations of NPPs. A case study using the 2019 M7.1 Ridgecrest earthquake can be for found in Appendix F.

## 5.1 NUMBER (TEMPORAL) TEST

The number test evaluates the temporal component of the forecast by checking the consistency of the forecasted number of events, $N$ with those observed in the forecast horizon, $N_{obs}$. Upper and lower quantiles are estimated using the empirical cumulative distribution from the repeat simulations, $F_N$,

$$\delta_1 = \mathbb{P}(N \geq N_{obs}) = 1 - F_N(N_{obs} - 1) \tag{7}$$
$$\delta_2 = \mathbb{P}(N \leq N_{obs}) = F_N(N_{obs}). \tag{8}$$

## 5.2 SPATIAL TEST

To evaluate the spatial component of the forecast, a test statistic aggregates the forecasted rates of earthquakes over a regular grid,

$$S = \left[\sum_{i=1}^{N} \log \hat{\lambda}(k_i)\right] N^{-1}, \tag{9}$$

where $\hat{\lambda}(k_i)$ is the approximate rate in the cell $k$ where the $i^{th}$ event is located. Upper and lower quantiles are estimated by comparing the observed statistic

$$S_{obs} = \left[\sum_{i=1}^{N_{obs}} \log \hat{\lambda}(k_i)\right] N_{obs}^{-1}, \tag{10}$$

with the empirical cumulative distribution of $S$ using the repeat simulations, $F_S$

$$\gamma_s = \mathbb{P}(S \leq S_{obs}) = F_S(S_{obs}). \tag{11}$$

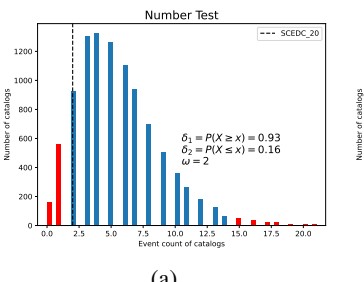 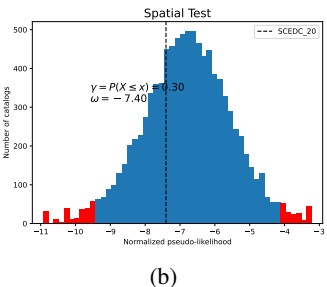 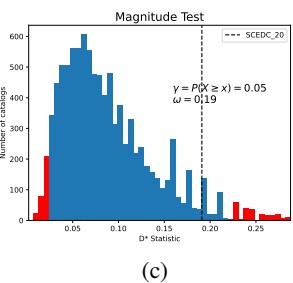

(a)              (b)              (c)

Figure 4: CSEP consistency tests on the ETAS model for the first day (01/01/2014) of the testing period in the `SCEDC` catalog. A total of 10,000 simulations are generated to compute empirical distributions of the test statistics for each of the three consistency tests: (a) Number test, (b) Spatial test, and (c) Magnitude test. The test fails if the observed statistic falls within the rejection region (red), defined by the 0.05 and 0.95 quantiles of the distribution.

The grid is constructed from $\{0.1°, 0.05°, 0.01°\}$ squares for `ComCat`, `SCEDC` and $\{$`QTM_Salton_Sea`, `QTM_SanJac`, `White`$\}$ respectively.

## 5.3 MAGNITUDE TEST

To evaluate the earthquake magnitude component of the forecast, a test statistic compares the histogram of a forecast's magnitudes $\Lambda^{(m)}$, against the mean histogram over all forecasts $\bar{\Lambda}^{(m)}$,

$$D = \sum_k \left( \log\left[\bar{\Lambda}^{(m)}(k) + 1\right] - \log\left[\Lambda^{(m)}(k) + 1\right]\right)^2, \tag{12}$$

where $\Lambda^{(m)}(k)$ and $\bar{\Lambda}^{(m)}(k)$ are the counts in the $k^{th}$ bin of the forecast and mean histograms, normalised to have the same total counts as the observed catalog. Upper and lower quantiles are estimated by comparing the observed statistic

$$D_{\text{obs}} = \sum_k \left( \log\left[\bar{\Lambda}^{(m)}(k) + 1\right] - \log\left[\Lambda^{(m)}_{\text{obs}}(k) + 1\right]\right)^2, \tag{13}$$

with the empirical distribution of $D$ using the repeat simulations, $F_D$

$$\gamma_m = \mathbb{P}(D \leq D_{\text{obs}}) = F_D(D_{\text{obs}}). \tag{14}$$

Histogram bins of size $\delta_m = 0.1$ are used across all datasets.

## 5.4 EVALUATING MULTIPLE FORECASTING PERIODS

Savran et al. (2020) describe how to assess a model's performance across the multiple days in the testing period. By construction, quantile scores over multiple periods should be uniformly distributed if the model is the data generator (Gneiting & Katzfuss, 2014). Therefore comparing quantile scores against standard uniform quantiles (y = x), highlights discrepancies between the observed data and the forecast. Additional statements can be made about over-prediction or under-prediction of each test statistic (quantile curves above/bellow y=x respectively). The Kolmogorov-Smirnov (KS) statistic then quantifies the degree of difference to the uniform distribution for each of the tests.

Further documentation of how to perform the CSEP evaluation procedure can be found on the platform, where we demonstrate the procedure for the ETAS model. Table 3 reports the benchmark performance scores taken from the quantile plots in Appendix D. The performance of ETAS is higher for the more typical higher magnitude catalogs such as `ComCat` and `SCEDC`, whereas it performs worse at the lower magnitude catalogs of `QTM_SanJac`, `QTM_SaltonSea` and `White`. Spatial prediction is consistently the best performing component of the ETAS forecast, whereas earthquake numbers are overpredicted by the model and earthquake magnitudes are generally not well predicted (Figure 9) . All results indicate significant room for improvement beyond the predictive performance of the ETAS model.

Table 3: CSEP consistency tests evaluate the calibration of all daily ETAS forecasts on EarthquakeNPP datasets. A test is performed at the $\alpha = 0.05$ significance level on each day in the testing period. The pass rate indicates the success of ETAS across all testing days. By construction quantile scores of the tests should be uniformly distributed if the model is the data generator. The KS-Statistic reports the difference of the quantile distribution to uniform, taken from the quantile plots in Appendix D.

| Dataset | Number Test | | Spatial Test | | Magnitude Test | |
|---|---|---|---|---|---|---|
| | Pass Rate | KS-Statistic | Pass Rate | KS-Statistic | Pass Rate | KS-Statistic |
| ComCat | 62.3% | 0.392 | 85.3% | 0.128 | 75.3% | 0.318 |
| SCEDC | 74.4% | 0.161 | 87.5% | 0.123 | 80.5% | 0.153 |
| QTM_SanJac | 59.2% | 0.461 | 96.7% | 0.145 | 66.2% | 0.406 |
| QTM_SaltonSea | 54.2% | 0.441 | 82.1% | 0.216 | 79.0% | 0.311 |
| White | 17.1% | 0.750 | 98.0% | 0.373 | 25.0% | 0.741 |

## 6 DISCUSSION AND CONCLUSION

We introduce the EarthquakeNPP datasets to facilitate the benchmarking of NPPs against a community-endorsed ETAS model for earthquake forecasting. These datasets cover various regions of California, representing typical forecasting zones and the data commonly available to forecast issuers. Several datasets use modern methods of detection, which enables the inclusion of much smaller magnitude earthquakes.

In a benchmarking experiment, we compared three NPP models against ETAS and a baseline Poisson process. None of the NPP models outperformed ETAS, indicating that current NPP implementations are not yet suitable for operational earthquake forecasting. ETAS explicitly defines how larger earthquake magnitudes increase the likelihood of future earthquakes in both time and space, using an empirical relationship derived from seminal observational studies (Utsu & Seki, 1955; Utsu, 1970). This use of magnitude information is shared across all competitive short-term earthquake forecasting models currently used operationally (Mizrahi et al., 2024a) or tested by CSEP (Taroni et al., 2018). The lack of a direct dependence on magnitudes in the current NPP implementations likely explains their relative under-performance compared to ETAS. Future implementations should exploit this additional feature for improved temporal and spatial performance. Encouragingly, the comparable temporal performance to ETAS without this additional feature suggests that incorporating magnitude dependence would enhance NPP performance beyond that of ETAS.

EarthquakeNPP supports the earthquake forecast evaluation procedure developed by the Collaboratory for the Study of Earthquake Predictability (CSEP). The procedure replicates how earthquakes forecasts are generated in an operational setting, requiring models to simulate many repeat event sequences over a day-long forecast horizon. Benchmark performance for the ETAS model enables future comparison of NPPs that are implemented for this procedure and enables their promotion to the fully prospective CSEP experiments. Notably, this procedure allows the evaluation of generative NPP models without explicit likelihoods (Yuan et al., 2023; Li et al.), by assessing their performance over the full trajectory of future events. Probabilistic seismic hazard analysis (PSHA) requires long-term prediction beyond the next-event (Ebrahimian et al., 2014; Gerstenberger et al., 2014), therefore this approach also offers stakeholders a more comprehensive understanding of earthquake hazard than metrics focused on predicting the next event (e.g. RMSE). The procedure also follows the recommendation by Shchur et al. (2021) to move away from next-event point prediction for NPPs.

The EarthquakeNPP datasets, available at https://anonymous.4open.science/r/EarthquakeNPP-2D51, provide a platform for future NPP developments to be benchmarked against these initial results. The platform is under ongoing development and in the future will see the direct comparison of emerging and other existing models models developed within the seismology community, as well as an expansion of datasets included to other seismically active global regions. Successful NPP models on these datasets, for both log-likelihood and CSEP metrics, will be directly impactful to stakeholders in seismology, ultimately enabling their integration into operational earthquake forecasting by government agencies.

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

# A EARTHQUAKE CATALOG DATA

## A.1 EARTHQUAKE CATALOG GENERATION

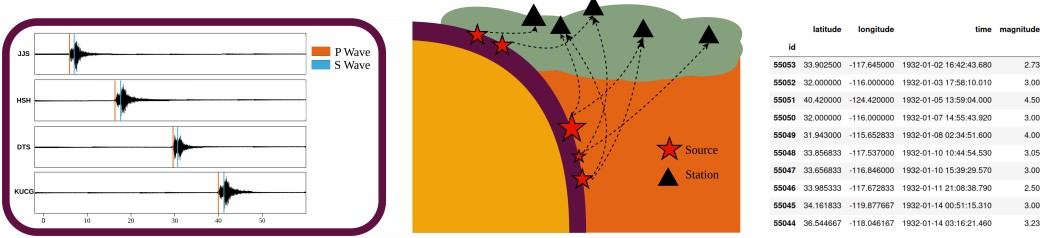

Figure 5: Generating an earthquake catalog involves several key steps: seismic phase picking, magnitude estimation, and the association and location of seismic sources. This process transforms raw waveform data recorded at seismic stations to locations, times, and magnitudes of earthquakes.

Data missingness, referred to in seismology as catalog (in)completeness, is the primary challenge faced with earthquake catalogs. It is an important and unavoidable feature, and is a result of how earthquakes are detected and characterised. Below, we briefly overview the process of generating an earthquake catalog to illustrate the data quality issues. In the subsequent section, we review catalog incompleteness and its potential impact on the performance and evaluation of forecast models.

**Seismometers and Seismic Networks.** A seismometer is an instrument that detects and records the vibrations caused by seismic waves (Stein & Wysession, 2009; Shearer, 2019). It consists of a sensor to detect ground motion and a recording system to log three-dimensional ground motion over time, typically vertical and horizontal velocities. Seismic networks, comprising multiple seismometers, monitor seismic activity at regional, national or global scales (see, e.g., (Woessner et al., 2010) and references therein). High-density networks with modern, sensitive equipment provide more detailed and accurate data, enhancing the ability to detect and analyse smaller and more distant earthquakes.

**From Waveforms to Phase Picking.** The process of converting raw continuous seismic waveforms into useful earthquake data begins with phase picking, which identifies the arrival times of the primary (P) and secondary (S) waves of an earthquake. Historically, this was done manually, but now automated algorithms, such as the STA/LTA algorithm, detect wave arrivals by analyzing signal amplitude changes (Allen, 1982). Recent algorithms, such as machine learning classifiers (e.g. Zhu & Beroza, 2019; Lapins et al., 2021) and template-matching (e.g. Ross et al., 2019), can process much higher volumes of data efficiently and are often able to detect events of much smaller magnitudes.

**Earthquake Association and Location** After phase picking, the next step is to associate phases from different seismometers with the same earthquake. Simple algorithms require at least four

phase arrivals to be detected on different stations within a short time interval to declare an event. Once phases are associated, location estimation determines the earthquake's hypocenter and origin time by minimizing travel-time residuals using linearized or global inversion algorithms (Thurber, 1985; Lomax et al., 2000). Given the potential for misidentified or mis-associated phase arrivals due to low signal-to-noise of small events or the near-simultaneous occurrence during very active aftershock sequences, an automated system typically first picks arrival times and determines a preliminary location, which is subsequently reviewed by a seismologist (e.g. Woessner et al., 2010, and references therein). Locations are typically reported as the geographical coordinates and depths where earthquakes first nucleated (hypocenters), although some catalogs report the centroid location, a central measure of the extended earthquake rupture.

**Earthquake Magnitude Calculation** The magnitude of an earthquake quantifies the energy released at the source and was originally defined in the seminal paper by Richter (1935). The original definition, now referred to as the local magnitude (ML), is calculated from the logarithm of the amplitude of waves recorded by seismometers. This scale, however, "saturates" at higher magnitudes, meaning it underestimates magnitudes for various reasons. This led to introduction of the moment magnitude scale (Mw) (Hanks & Kanamori, 1979), which computes the magnitude based on the estimated seismic moment $M_0$, which can be related to the physical rupture process via

$$M_0 = \text{rigidity} \times \text{rupture area} \times \text{slip}, \tag{15}$$

where rigidity is a mechanical property of the rock along the fault, rupture area is the area of the fault that slipped, and slip is the distance the fault moved. Mw is determined seismologically via a spectral fitting process to the earthquake waveforms. In practice, it can be challenging to use a single magnitude scale for a broad range of magnitudes, therefore a range of scales may be present within a single catalog, and approximate magnitude conversion equations may be used to homogenize the scales (e.g. Herrmann & Marzocchi, 2021, and references therein).

## A.2 EARTHQUAKE CATALOG COMPLETENESS

All of the EarthquakeNPP datasets are made publicly available by their respective data centers in raw format. However, constructing a suitable retrospective forecasting experiment from this raw data requires appropriate pre-processing. This typically involves truncating the dataset above a magnitude threshold $M_{\text{cut}}$ and within a target spatial region to address incomplete data, known as catalog completeness $M_c$ (e.g., Mignan et al., 2011; Mignan & Woessner, 2012).

There are several reasons why an earthquake may not be detected by a seismic network. Small events may be indistinguishable from noise at a single station, or insufficiently corroborated across multiple stations. Another significant cause of missing events occurs during the aftershock sequence of large earthquakes, when the seismicty rate is high (Kagan & Knopoff, 1987; Hainzl, 2022). Human or algorithmic detection abilities are hampered when numerous events occur in quick succession, e.g. when phase arrivals of different events overlap at different stations or the amplitudes of small events are swamped by those of large events. Since catalog incompleteness increases for lower magnitude events, typically the task is to find the value $M_c$ above which there is approximately $100\%$ detection probability. Choosing a truncation threshold $M_{\text{cut}}$ that is too high removes usable data. Where NPPs have demonstrated an ability to perform well with incomplete data (Stockman et al., 2023), typically a threshold below the completeness biases classical models such as ETAS (Seif et al., 2017). Seismologists often investigate the biases of different magnitude thresholds by performing repeat forecasting experiments for different thresholds (e.g. Mancini et al., 2022; Stockman et al., 2023), which we also facilitate in our datasets.

Typically $M_c$ is determined by comparing the raw earthquake catalog to the Gutenberg-Richter law (Gutenberg & Richter, 1936), which states that the distribution of earthquake magnitudes follows an exponential probability density function

$$f_{GR}(m) = \beta e^{\beta(m - M_c)} \quad : m \geq M_c. \tag{16}$$

where $\beta$ is a rate parameter related to the b-value by $\beta = b \log 10$. Histogram-based approaches, such as the simple Maximum Curvature method (Wiemer & Wyss, 2000) as well as many others (e.g. Herrmann & Marzocchi, 2021, and references therein), identify the magnitude at which the observed catalog deviates from this law, indicating incompleteness (See Figure 6b).

In practice, catalog completeness varies in both time and space $M_c(t, \mathbf{x})$ (e.g. Schorlemmer & Woessner, 2008). During aftershock sequences, $M_c(t)$ can be very high (e.g., Agnew, 2015; Hainzl, 2016b) (See Figure 6a). Thresholding at the maximum value might remove too much data. Instead, modelers either omit particularly incomplete periods during training and testing (Kagan, 1991; Hainzl et al., 2008), model the incompleteness itself (Helmstetter et al., 2006; Werner et al., 2011; Omi et al., 2014; Hainzl, 2016a;b; Mizrahi et al., 2021; Hainzl, 2022), or accept known biases from disregarding this issue (Sornette & Werner, 2005). Spatially, catalogs are less complete farther from the seismic network (Mignan et al., 2011), so the spatial region can be constrained to remove outer, more incomplete areas (See Figure 6c).

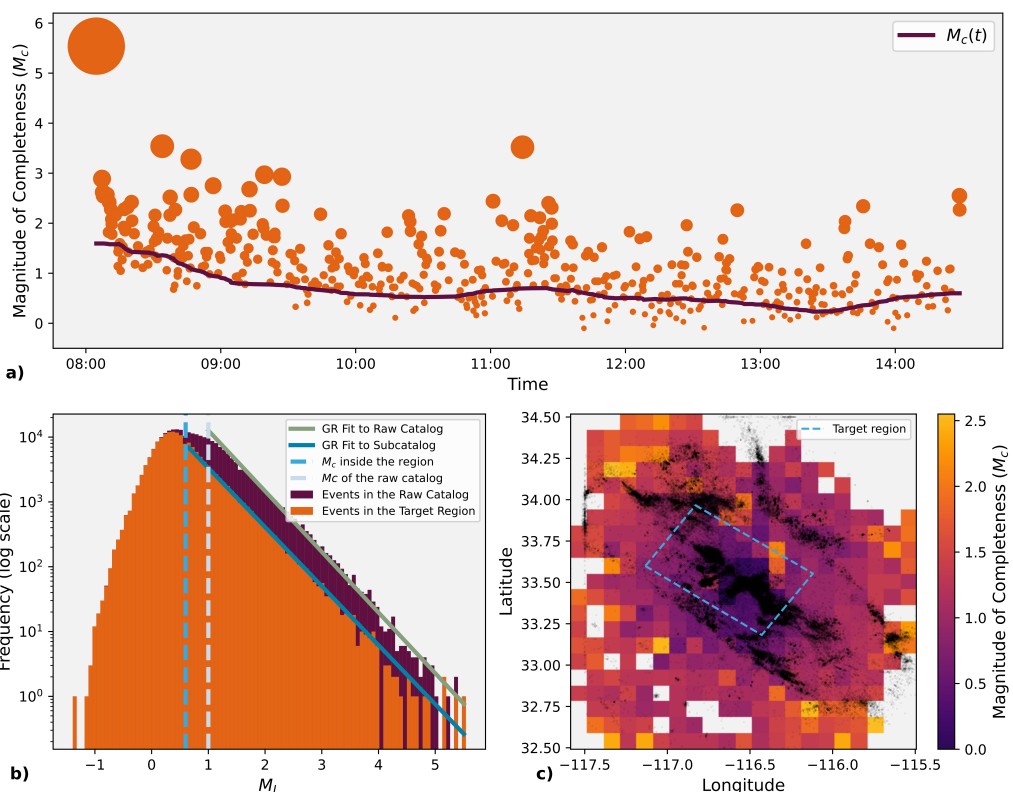

Figure 6: **a)** the June 10, 2016 Mw5.2 Borrego Springs earthquake and aftershocks, which occurred on the San Jacinto fault zone and is recorded in the WHITE catalog. An estimate of the magnitude of completeness $M_c(t)$ over time using the Maximum Curvature method reveals more incompleteness immediately following the large earthquake. **b)** magnitude-frequency histograms reveal that truncating the raw WHITE catalog to inside the target region decreases $M_c$. Each histogram is fit to the Gutenberg-Richter (GR) law and an estimate of $M_c$ for each catalog occurs where the histogram deviates from the (GR) line. **c)** An estimate of $M_c$ for gridded regions of the San Jacinto fault zone, using the raw WHITE catalog.

## B ADDITIONAL DATASETS

Beyond the official EarthquakeNPP datasets, we include 3 further datasets that either provide additional scientific insight or continuity from previous benchmarking works.

**Synthetic ETAS Catalogs.** We simulate a synthetic catalog using the ETAS model with parameters estimated from ComCat, at $M_c$ 2.5, within the same California region. A second catalog emulates the time-varying data-missingness present in observational catalogs by removing events using the time-dependent formula from Page et al. (2016),

$$M_c(M, t) = M/2 - 0.25 - \log_{10}(t), \tag{17}$$

Table 4: Summary of additional datasets, including: magnitude threshold ($M_c$), number of training events, and number of testing events. The chronological partitioning of training, validation, and testing periods is also detailed. An auxiliary (burn-in) period begins from the "**Start**" date, followed by the respective starts of the training, validation, and testing periods. All dates are given as 00:00 UTC on January 1st, unless noted (* refers to 00:00 UTC on January 17th).

| Catalog | $M_c$ | Start-Train-Val-Test-End | Train Events | Test Events |
|---|---|---|---|---|
| ETAS | 2.5 | 1971-1981-1998-2007-2020* | 117,550 | 43,327 |
| ETAS_incomplete | 2.5 | 1971-1981-1998-2007-2020* | 115,115 | 42,932 |
| Japan_Deprecated | 2.5 | 1990-1992-2007-2011-2020 | 22,213 | 15,368 |

where $M$ is the mainshock magnitude. Events below this threshold are removed using mainshocks of Mw 5.2 and above. The inclusion of these datasets allows us to test whether NPPs are inhibited by data missingness to the same extent that ETAS is.

**Deprecated Catalog of Japan.** To provide continuity from the previous benchmarking for NPPs on earthquakes, we also provide results on the Japanese dataset from Chen et al. (2021), however with a chronological train-test split and without removing any supposed outlier events. To reflect our recommendation not to use this dataset in any future benchmarking following the dataset completeness issues mentioned above, we name this dataset Japan_Deprecated.

Figures 7 and 8 report the temporal and spatial log-likelihood scores of all the benchmarked models on additional datasets. On synthetic data generated by the ETAS model the performance of NPPs mirrors the results on the observational data (Figures 2 and 3). The performance of NPPs is more comparable to ETAS in terms of temporal log-likelihood however they cannot capture the distribution of earthquake locations. Change in temporal performance of models between the ETAS and ETAS_incomplete datasets reveal each model's robustness to the missing data typically present in earthquake catalogs (See section A.2). Auto-STPP and ETAS reduce in performance upon the removal earthquakes during aftershock sequences, whereas DeepSTPP and NSTPP maintain the same performance indicating a robustness to the data missingness.

On the Japan_Deprecated dataset, whilst ETAS remains the best performing model for spatial prediction, for temporal prediction it performs comparably to NSTPP and is even marginally outperformed by DeepSTPP. This performance can be attributed to the data completeness issues of the Japan_Deprecated dataset (see section 1.1), where the test period is missing all earthquakes bellow magnitude 4.0.

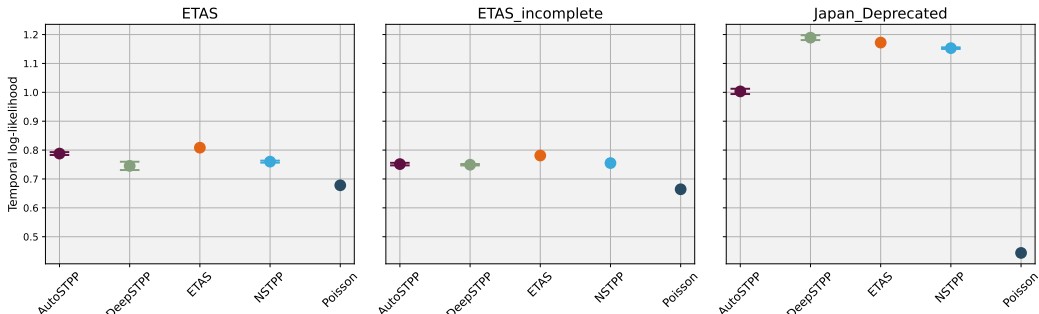

Figure 7: Test temporal log-likelihood scores for all the spatio-temporal point process models on each of the additional datasets. Error bars of the mean and standard deviation are constructed for the NPPs using three repeat runs.

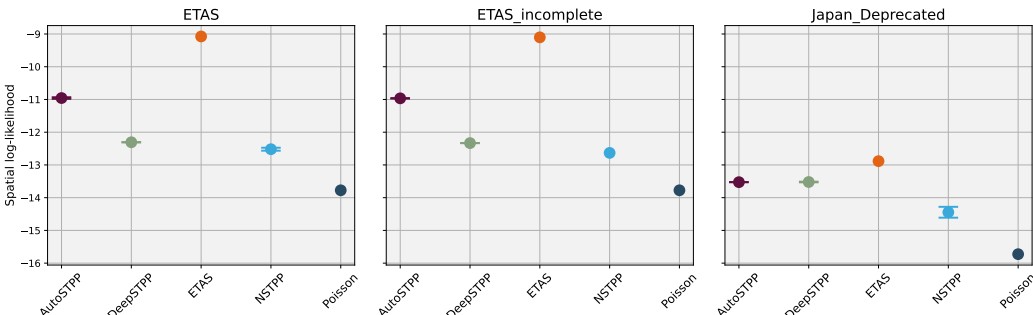

Figure 8: Test spatial log-likelihood scores for all the spatio-temporal point process models on each of the additional datasets. Error bars of the mean and standard deviation are constructed for the NPPs using three repeat runs.

## C  COMPUTATIONAL EFFICIENCY

### C.1  TRAINING

Table 5 reports the training times for each model across all datasets. We ran all the NPP models using a HPC node with Nvidia Ampere GPU with 4x Nvidia A100 40GB SXM "Ampere" GPUs and AMD EPYC 7543P 32-Core Processor "Milan" CPU using torch==1.12.0 and cuda==11.3.

| Dataset | # Training Events | ETAS | Deep-STPP | AutoSTPP | NSTPP | Poisson |
|---|---|---|---|---|---|---|
| ComCat | 79,037 | 08:59:04 | 00:15:35 | 01:34:09 | 3 days, 05:10:17 | <1 second |
| QTM_SaltonSea | 44,042 | 07:28:28 | 00:26:46 | 01:45:34 | 2 days, 00:26:45 | <1 second |
| QTM_SanJac | 18,664 | 00:32:40 | 00:09:31 | 00:37:03 | 1 day, 22:06:33 | <1 second |
| SCEDC_20 | 128,265 | 13:42:30 | 00:38:10 | 02:54:51 | 3 days, 02:20:40 | <1 second |
| SCEDC_25 | 43,221 | 03:09:14 | 00:09:34 | 00:56:05 | 2 days, 16:33:55 | <1 second |
| SCEDC_30 | 12,426 | 00:42:25 | 00:02:44 | 00:16:01 | 1 day, 16:39:04 | <1 second |
| White | 38,556 | 03:55:40 | 00:08:21 | 01:10:51 | 2 days, 01:03:57 | <1 second |
| Japan_Deprecated | 22,213 | 06:09:08 | 00:13:45 | 01:02:07 | 2 days, 05:32:03 | <1 second |
| ETAS | 117,550 | 00:33:25 | 00:15:24 | 01:10:22 | 3 days, 03:09:17 | <1 second |
| ETAS_incomplete | 115,115 | 00:35:14 | 00:15:29 | 01:09:43 | 3 days, 11:39:51 | <1 second |

Table 5: Training times for each model across all datasets, including the number of training events. Times are formatted as HH:MM:SS, with days included for durations exceeding 24 hours. The Poisson model consistently requires less than 1 second.

**ETAS** training scales $\mathcal{O}(n^2)$ with the total number of events, since for every event a contribution to the intensity function is computed from a summation over all previous events. This scaling, coupled with the lack of parallelization in the current implementation, results in long training times for larger datasets. Poorer scaling will likely hinder **ETAS** if dataset sizes continue to grow in the future (Stockman et al., 2024).

Encouragingly, both **Deep-STPP** and **AutoSTPP** are significantly faster to train due to GPU acceleration and their use of a sliding window of the most recent $k = 20$ events. While exact complexity analyses are not provided in Zhou et al. (2022) or Zhou & Yu (2024), we can infer that **Deep-STPP** likely scales as $\mathcal{O}(kn)$ since it benefits from a closed-form expression for the likelihood. **AutoSTPP**,

though requiring automatic integration to compute the likelihood, still scales with $\mathcal{O}(kn)$ because the additional integration cost does not affect the overall scaling.

**NSTPP**, on the other hand, incurs significant training costs, rendering it impractical for real-time forecasting. Unlike the sliding window mechanism used in **Deep-STPP** and **AutoSTPP**, **NSTPP** partitions the event sequence into fixed time intervals, leading to sequences that are much longer than the $k = 20$ events used by the other models (as shown in Figure 11 of Chen et al. (2021)). Furthermore, solving an ODE for each event time adds a significant computational burden, even with the use of their faster attentive CNF architecture.

## C.2 SIMULATION

Real-time earthquake forecasting and CSEP model evaluation require simulating many repeat sequences (at least 10,000 for adequate distributional coverage) over the forecasting horizon. While ETAS training scales as $\mathcal{O}(n^2)$ with the number of training events, its simulation scales more efficiently at $\mathcal{O}(n \log n)$. This improved scaling is due to its equivalent formulation as a Hawkes branching process (see Section 2.2). Both Deep-STPP and AutoSTPP are also based on Hawkes processes, which theoretically allows for fast simulation. However, as these models currently only have an intensity function implementation, simulating events would require a slower thinning procedure (Ogata, 1981), limiting their simulation efficiency. In contrast, NSTPP benefits from fast simulation, owing to its design using continuous-time normalizing flows. Events can be generated by passing samples from a base distribution through learned transformations, resulting in a much faster simulation process.

## D CSEP CONSISTENCY TESTS

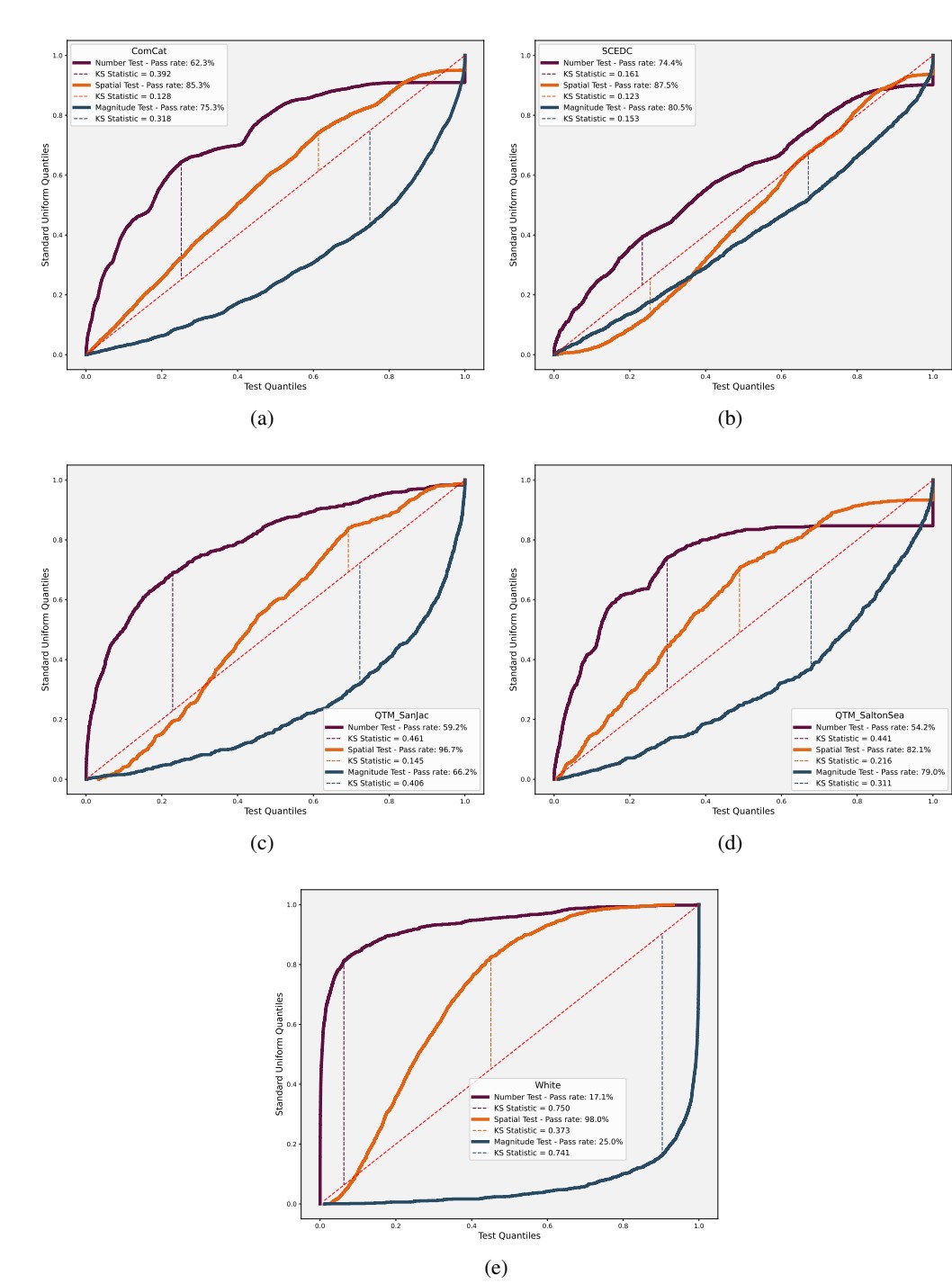

Figure 9: Quantile-quantile plots showing the calibration of all daily ETAS forecasts on a) `ComCat`,b) `SCEDC`, c) `QTM_San_Jac`, d) `QTM_Salton_Sea`, e) `White`. By construction quantile scores over multiple periods should be uniformly distributed if the model is the data generator. Comparing quantile scores against standard uniform quantiles ($y = x$), highlights discrepancies between the observed data and the forecast. Pass rates of each test are indicated in the legend. The Kolmogorov-Smirnov statistic, quantifies the degree of difference to the uniform distribution.

## E  FURTHER DATASET FIGURES

### E.1  COMCAT

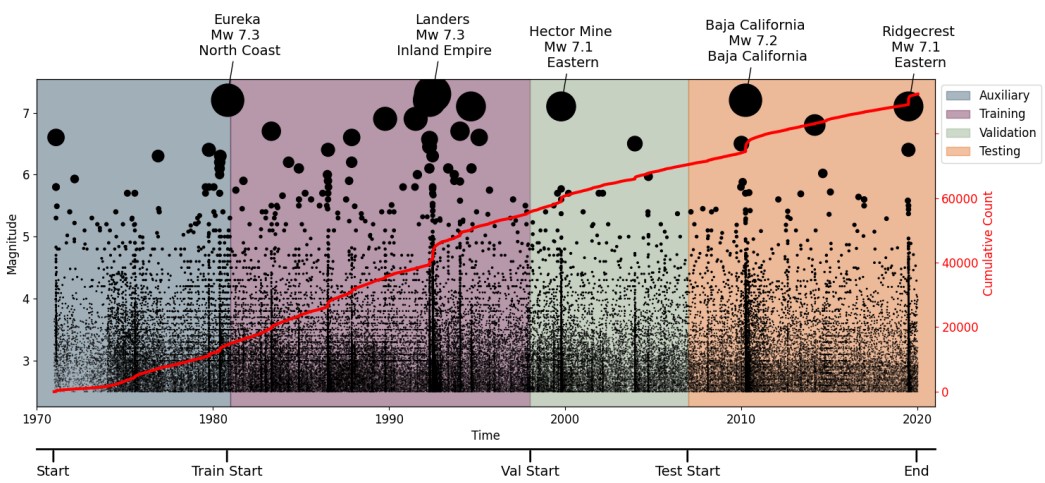

Figure 10: Times and magnitudes of events in the ComCat dataset (with key events labeled). The size of the points are plotted on a log scale corresponding to Mw. Auxiliary, training, validation and testing periods are indicated by colour and a further cumulative count of events is indicated in red.

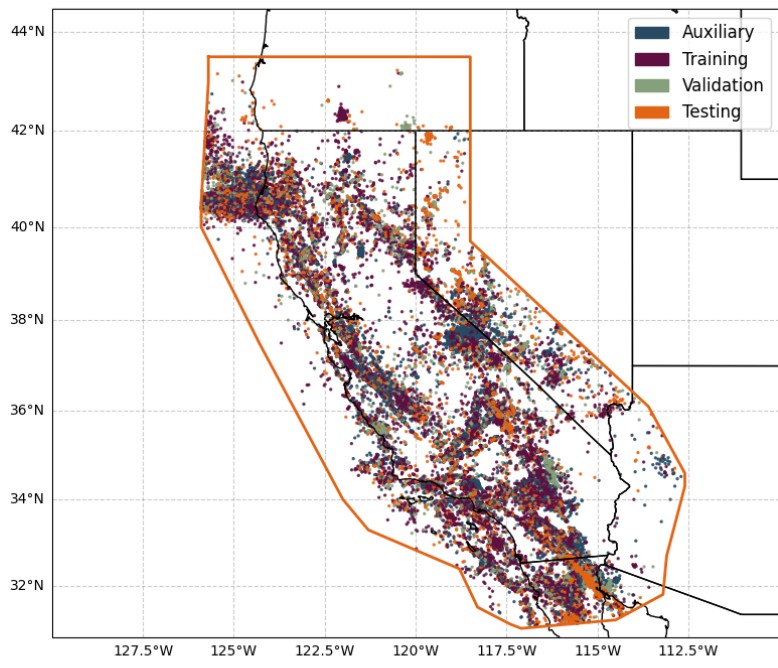

Figure 11: Locations of events in the ComCat dataset, labeled by their partition into auxiliary, training, validation and testing periods.

## E.2 SCEDC

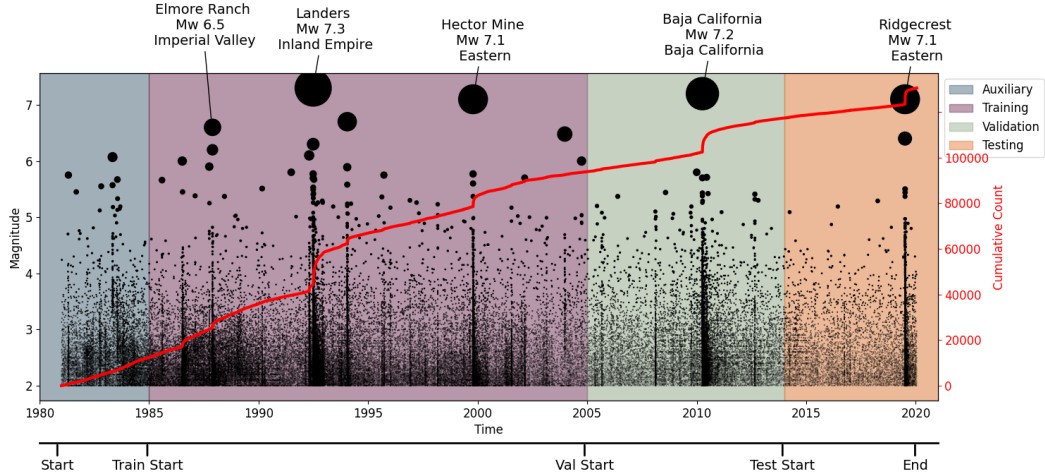

Figure 12: Times and magnitudes of events in the SCEDC dataset (with key events labeled). The size of the points are plotted on a log scale corresponding to Mw. Auxiliary, training, validation and testing periods are indicated by colour and a further cumulative count of events is indicated in red.

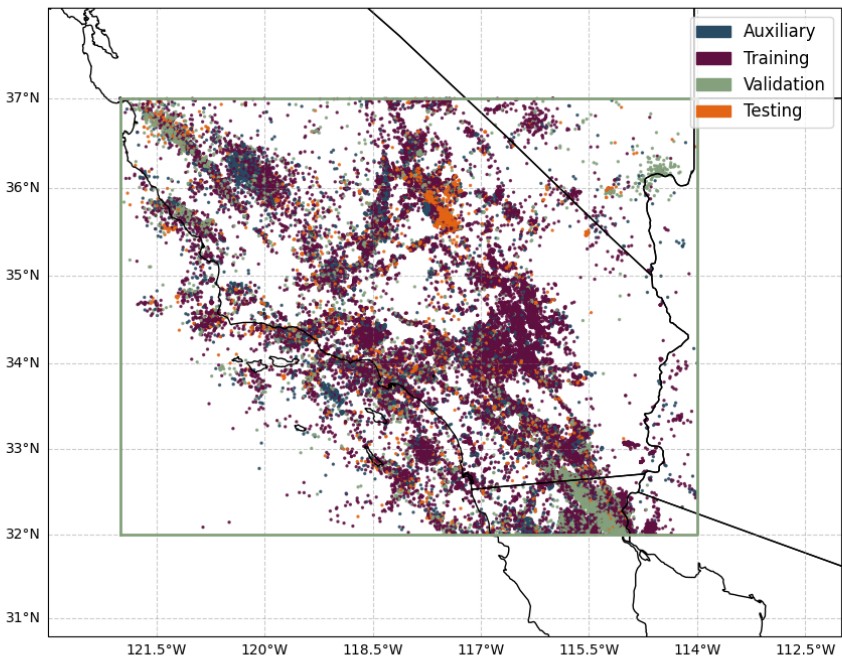

Figure 13: Locations of events in the SCEDC dataset, labeled by their partition into auxiliary, training, validation and testing periods.

### E.3 WHITE

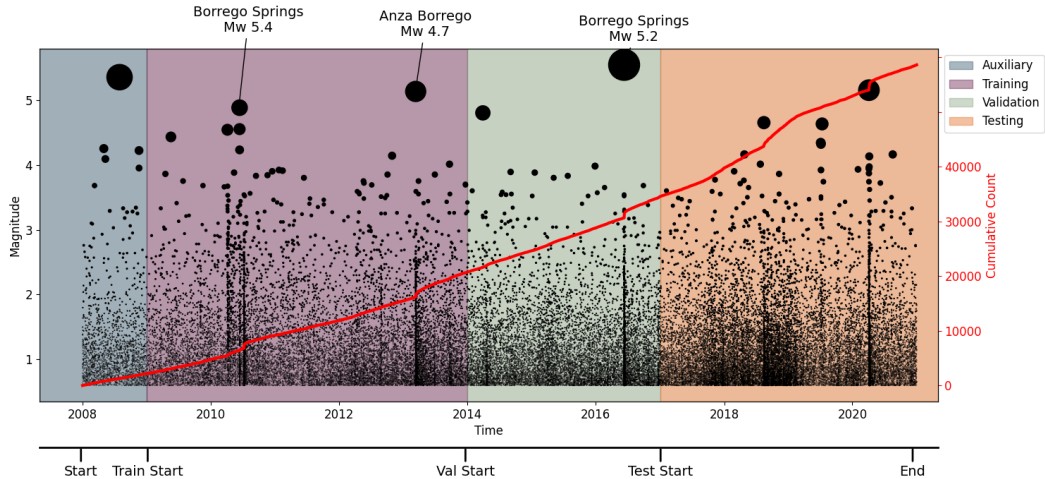

Figure 14: Times and magnitudes of events in the `White` dataset (with key events labeled). The size of the points are plotted on a log scale corresponding to Mw. Auxiliary, training, validation and testing periods are indicated by colour and a further cumulative count of events is indicated in red.

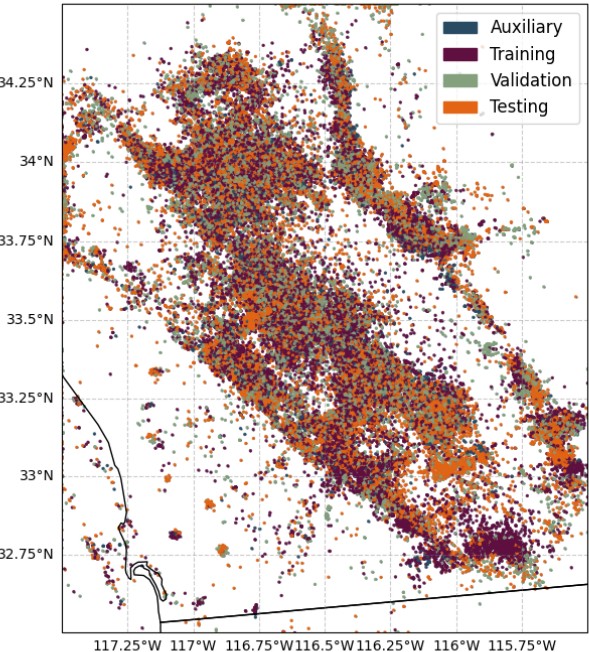

Figure 15: Locations of events in the `White` dataset, labeled by their partition into auxiliary, training, validation and testing periods.

### E.4  QTM_SANJAC

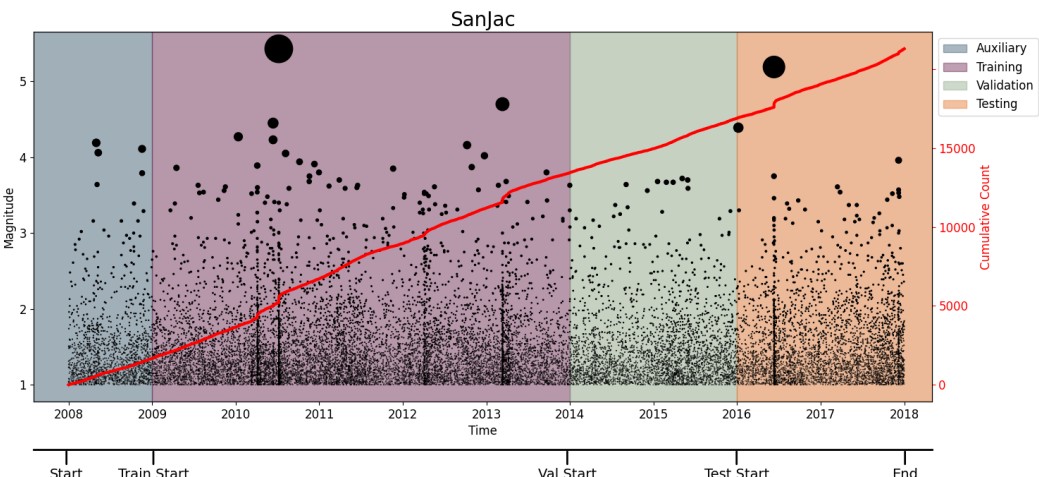

Figure 16: Times and magnitudes of events in the `QTM_SanJac` dataset. The size of the points are plotted on a log scale corresponding to Mw. Auxiliary, training, validation and testing periods are indicated by colour and a further cumulative count of events is indicated in red.

### E.5  QTM_SALTONSEA

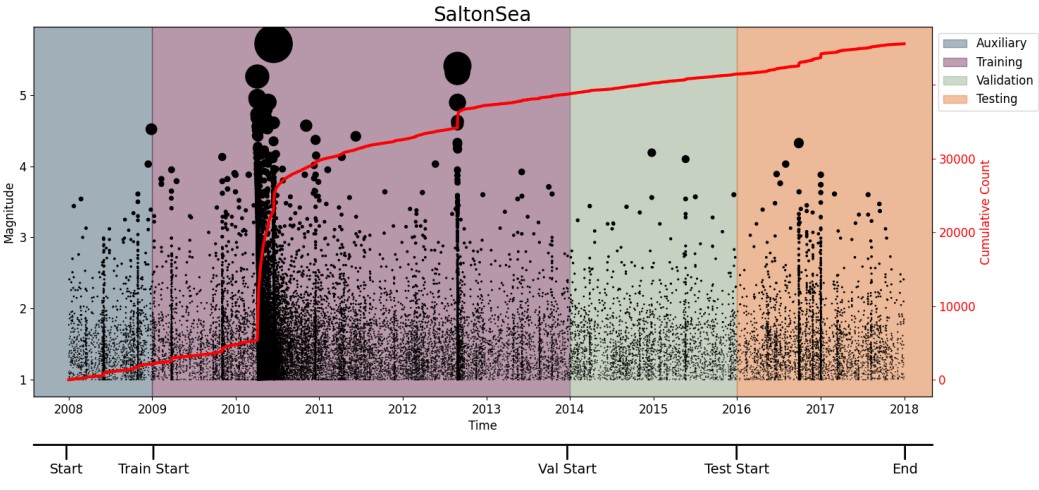

Figure 17: Times and magnitudes of events in the `QTM_SaltonSea` dataset. The size of the points are plotted on a log scale corresponding to Mw. Auxiliary, training, validation and testing periods are indicated by colour and a further cumulative count of events is indicated in red.

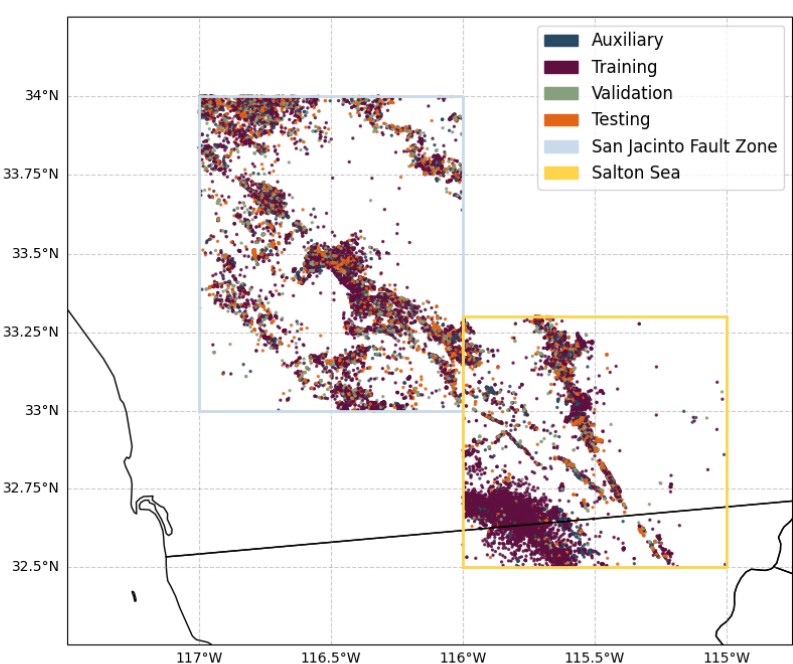

Figure 18: Locations of events in the `QTM_SanJac` and `QTM_SaltonSea` datasets, labeled by their partition into auxiliary, training, validation and testing periods.

## F 2019 M7.1 RIDGECREST EARTHQUAKE CASE STUDY

The 2019 Ridgecrest earthquake sequence (Figure 19) was the most powerful seismic event to strike Southern California in the past 20 years. Centered near the town of Ridgecrest and the Naval Air Weapons Station China Lake, the sequence began with a magnitude 6.4 foreshock on July 4, 2019, at 17:33:49 UTC, followed by a more powerful magnitude 7.1 mainshock on July 6, 2019, at 03:19:53 UTC, both along the Eastern California Shear Zone. The earthquakes caused widespread surface rupture, with displacements along multiple faults, and triggered tens of thousands of aftershocks over the following months.

The impacts of the sequence were substantial. In Ridgecrest and surrounding areas, the shaking damaged homes, businesses, and infrastructure, including roads, water lines, and electrical systems. Fires broke out due to ruptured gas lines, exacerbating the destruction. The mainshock caused over $1 billion in damages, including significant damage to the China Lake Naval facility, which was temporarily evacuated and declared "not mission capable." Despite the severity of the shaking, no fatalities occurred, largely due to the remote location and earthquake-resistant construction in the region.

Using the CSEP evaluation procedure (Section 5), we isolate the performance of a model during the sequence to identify its strengths and weaknesses. Here, we apply this analysis to the ETAS model, illustrating how similar evaluations can be conducted for future implementations of NPPs or other machine learning-based models.

Figure 20 presents the results of the Number Test over the initial days of the sequence. ETAS forecasts consistently underestimate the number of aftershocks during the most seismically active phase of the sequence. It is only 4 days after the M7.1 Ridgecrest mainshock, that ETAS begins to provide accurate earthquake rate forecasts. Figure 21a shows the spatial forecast for the day after the M7.1

mainshock. While the forecasts successfully trace the likely aftershock zone, they are over-dispersed and exhibit an isotropic distribution around a centroid. This prevents the forecasts from accurately capturing the elongated and clustered orientation of seismicity along the fault, causing it to fail the Spatial Test (Figure 21b).

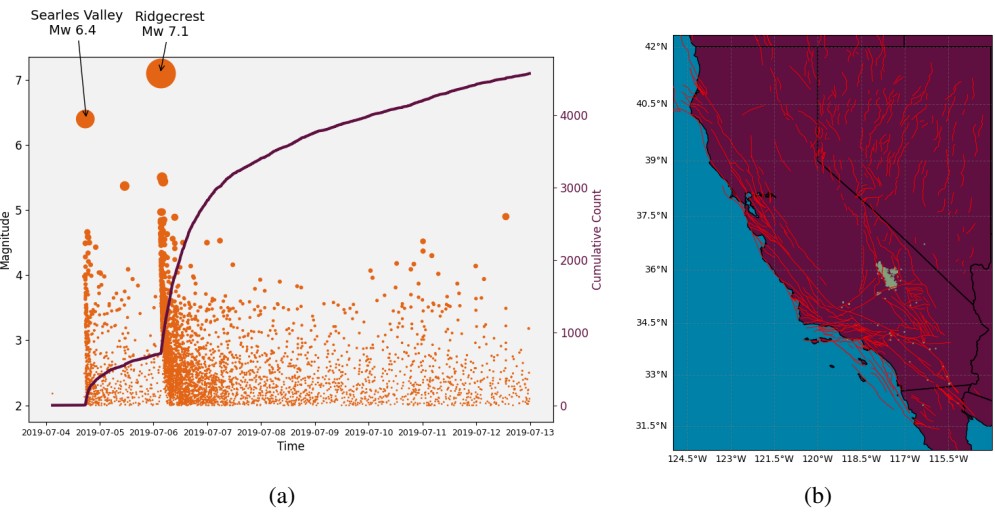

(a)                                    (b)

Figure 19: The 2019 Ridgecrest earthquake sequence began with the M6.4 Searles Valley foreshock on July 4, 2019, at 17:33:49 UTC, followed by the M7.1 Ridgecrest mainshock on July 6, 2019, at 03:19:53 UTC. (a) The times and magnitudes of events in the sequence. (b) Events in the sequence are plotted on a map of modeled faults in California.

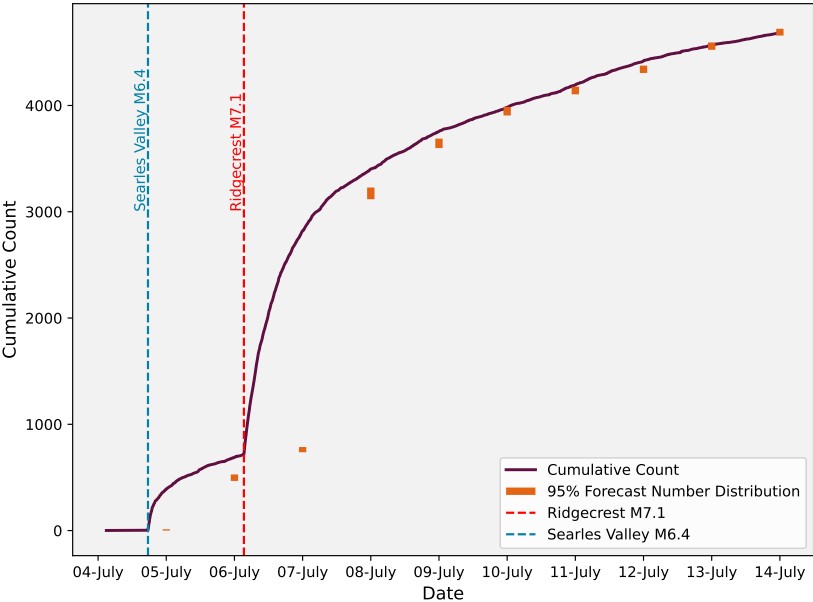

Figure 20: Forecasted earthquake number distribution using the ETAS model during the first 10 days of the Ridgecrest earthquake sequence. The number distributions are generated through 10,000 repeat simulations of earthquake sequences from the beginning of the day. The 95% confidence interval of the forecasted counts, generated at the start of each day, is compared to the observed number of events recorded by the end of the day.

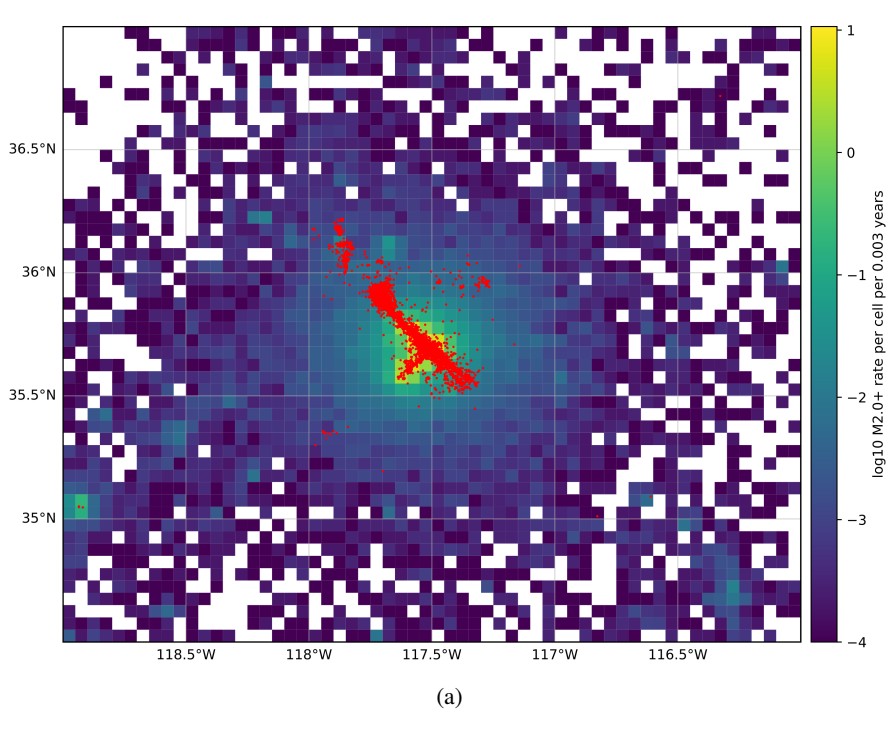

(a)

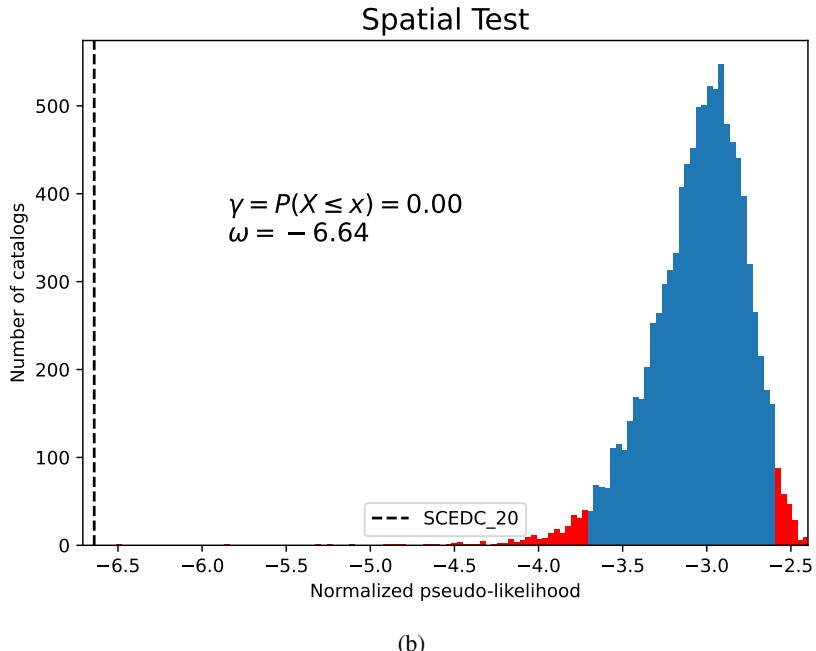

(b)

Figure 21: (a) The forecasted rates of earthquakes on July 7 (the day after the M7.1 Ridgecrest earthquake) using the ETAS model. Rates are calculated through 10,000 repeat simulations of earthquake sequences from the beginning of the day, which are aggregated to estimate a rate per spatial grid cell. In red are the observed earthquakes that occurred that day. (b) The results of the Spatial Test for July 7. Since the observed statistic is well outside the forecast distribution, the test is failed.

# G    ERROR DISTRIBUTIONS & NEXT-EVENT METRICS

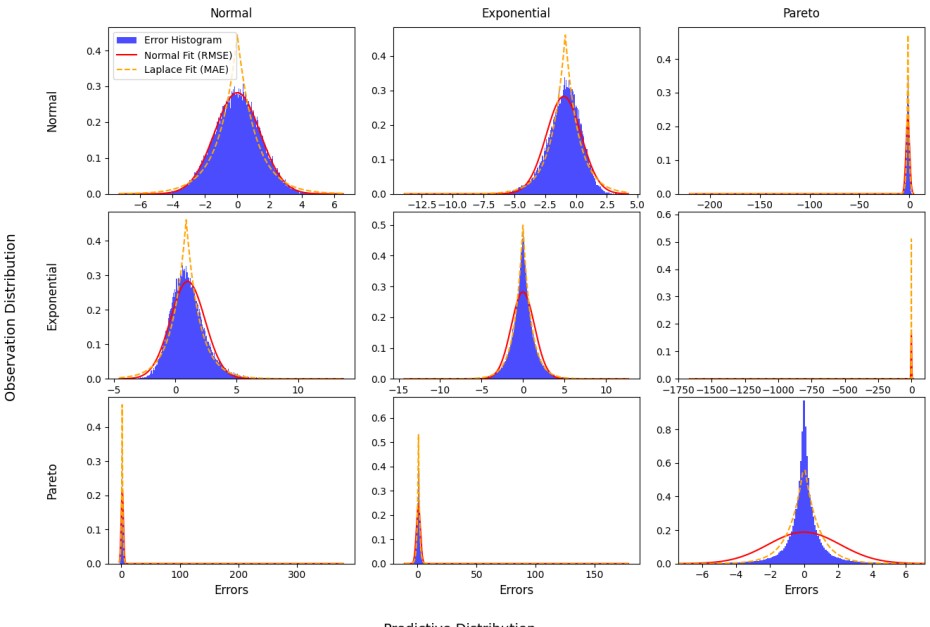

Figure 22: The distribution of errors ($Y_{\text{obs}} - Y_{\text{pred}}$) for the Normal$(0, 1)$, Exponential$(1)$, and Pareto$(2.5)$ distributions. Maximum likelihood estimation is used to fit Normal and Laplace distributions to each error histogram. Normal errors (Normal $\times$ Normal) are best approximated by the Root Mean Square Error (RMSE), while Laplacian errors (Exponential $\times$ Exponential) are best approximated by the Mean Absolute Error (MAE). However, neither RMSE nor MAE effectively capture the errors for the heavy-tailed Pareto distribution.

