# OpenReview forum: "EarthquakeNPP: Benchmark Datasets for Earthquake Forecasting with Neural Point Processes"
_ICLR.cc/2025/Conference — Submitted to ICLR 2025_

### Official Review · Reviewer_nWWY · 2024-10-29

**Soundness:** 2
**Presentation:** 3
**Contribution:** 2
**Rating:** 6
**Confidence:** 2

**Summary:**

This paper constructs a benchmark dataset for earthquake forecasting using machine learning. It constructs datasets of various periods, regions, and sizes and provides an evaluation method for them. It presents the results of predicting the dataset using the epidemic-type aftershock sequence (ETAS), a classical prediction method, and three Neural Point Processes (NPPs) models.

**Strengths:**

- Provides data over a more extended period of time, a wider area, and a wider variety of magnitude than existing benchmark datasets.
- The process of constructing the dataset was specifically explained. (Data period, scale, characteristics of each institution's data and etc.)
- The differences between the existing deep learning data split setting and the existing method were well explained.
- Various seismic domain-related experiments were conducted, such as number test, spatial test, and magnitude test.

**Weaknesses:**

- An analysis should be added on which period, time, and location the NPPs models predicted accurately/inaccurately (Qualitative results).

- It would be better to provide specific experimental results or references rather than “Since the NPPs lack any direct dependence on magnitudes” as the reason why NPPs perform worse than ETAS.

- If a new methodology appropriate for the task is proposed through the above analysis, it would be a better study.

- It would be good to organize the advantages more useful in the machine learning community than the existing benchmark datasets and present them in a table.

- Adding real-world earthquake forecasting and response scenarios using NPPs will improve applicability.

**Questions:**

- Can you explain the specific benefits of using classic ETAS compared to NPPs or other machine learning methods? For example, how do they differ in terms of computational load, efficiency, and cost? It would also be helpful to include the motivation for why solving this task with machine learning methods could be beneficial.
- Is there a technical reason for applying only NPPs among the various machine learning methodologies? It would be helpful if you could justify the choice of NPPs by providing a direct technical comparison with other machine learning methods.
- Have there been any previous studies, or do you have any experimental results, that applied other machine learning methods (e.g., spatio-temporal methods) besides NPPs?

---

> ### Author Response · Authors · 2024-11-19
> **Response to Reviewer nWWY (1/3)**
>
> We thank the reviewer for taking the time to read our paper and for such valuable feedback. Below we provide our responses in a point-by-point manner:
>
> > An analysis should be added on which period, time, and location the NPPs models predicted accurately/inaccurately (Qualitative results).
>
> We completely agree that this type of qualitative comparison would significantly improve the understanding of our experimental results. We chose to concentrate our efforts in this work on the presentation of datasets and seismology concepts to a machine learning audience, leaving little space for the further analysis that should be done interpreting model performance (this is often the subject of a whole paper in seismology e.g. Stockman et al., 2023). However, your suggestion (as well as your later point about providing a real-world forecasting situation) has prompted us to provide a forecasting "case study" that can be used as an example for how to quantitatively & qualitatively interpret forecasting results in a more fine grain manner.
>
> Whilst the log-likelihood metric can effectively summarise the "information gain" from one model over another (average properties of performance across the whole dataset), this is at the cost of fine grain interpretability (such as the qualitative analysis you are suggesting). To enhance interpretability (amongst other reasons) we added the CSEP evaluation procedure to the benchmark datasets. These metrics offer the chance to evaluate the models as if they were forecasting in a real-time manner, making qualitative statements about over/under prediction per day of the forecast or the clustering of the spatial forecasts. A new section of the appendix (titled "2019 M7.2 Ridgecrest Earthquake Case Study") demonstrates how the CSEP metrics can be used to analyse a model's performance during the 2019 Ridgecrest earthquake sequence (the most recent challenging real-time forecasting scenario for Californian agencies). Whilst this demonstration is only for the ETAS model, it will encourage future NPP works to perform such qualitative analysis using our benchmark datasets.
>
> > It would be better to provide specific experimental results or references rather than “Since the NPPs lack any direct dependence on magnitudes” as the reason why NPPs perform worse than ETAS. If a new methodology appropriate for the task is proposed through the above analysis, it would be a better study.
>
> Producing these experimental results would require implementing magnitude dependence for all the NPPs in our benchmark. This would require modeling choices to be made for every single model, including a discussion and description of these choices. Whilst using our benchmark to find "new methodology" is absolutely a next-step for this work, we believe this is well beyond the scope and distracts from our presentation of datasets and seismological concepts to a machine learning audience.
>
> Whilst we don't have any direct experimental evidence for this speculation, it is well accepted in seismology that the magnitudes of past earthquakes significantly affects the rate of future earthquakes. Seminal empirical studies (Utsu & Seki, 1955; Utsu, 1970) show this, and as a result, all competitive short-term earthquake forecasting models use this magnitude information (Taroni et al., 2018; Mizrahi et al., 2024).
>
> It was our hope that the references we cited in line 516 of the original manuscript provided enough evidence that magnitude information is a significant predictor of future earthquake rates. To emphasise this point further, we have added a further statement:
>
> "A consequence of these important observational studies is that all competitive short-term earthquake forecasting models used operationally (Mizrahi et al., 2024) or tested by CSEP (Taroni et al., 2018) use magnitude information for their forecasts."
>
> > It would be good to organize the advantages more useful in the machine learning community than the existing benchmark datasets and present them in a table.
>
> Thank you for this suggestion. We agree that a table would very effectively summarise the improvements of our datasets over the existing one. We shall include the following table:
>
>
> | Dataset       | Chronological Training/Test Splits | Complete Timespan | Complete Magnitudes | Used by Local Agencies |
> |---------------|-----------------------|----------------------------|----------------------|-------------------------|
> | Chen et al. (2021) Dataset   | ❌                   | ❌                         | ❌                   | ❌
> | EarthquakeNPP Datasets  | ✔️                   | ✔️                         | ✔️                   | ✔️       |

---

> ### Author Response · Authors · 2024-11-19
> **Response to Reviewer nWWY (2/3)**
>
> > Adding real-world earthquake forecasting and response scenarios using NPPs will improve applicability.
>
> Thank you for this suggestion. We agree that adding a real-time "case study" demonstrates how a model using our benchmark would be applied in a forecasting and response scenario (as well as offering an opportunity for more fine grain interpretation of the quality of the forecasts). As mentioned above, we have added a new section to the appendix titled "Ridgecrest Case Study". The 2019 Ridgecrest Earthquake was the most powerful earthquake to strike California in the past 25 years and cost an estimated $5.3 billion in damage. We believe your suggestion also provides much more context to the importance improving real-time forecasting with this case-study.
>
>
> > Can you explain the specific benefits of using classic ETAS compared to NPPs or other machine learning methods? For example, how do they differ in terms of computational load, efficiency, and cost? It would also be helpful to include the motivation for why solving this task with machine learning methods could be beneficial.
>
> Thank you for this suggestion. We agree that whilst we focused on motivating our datasets, we failed to adequately motivate why NPPs would/could be beneficial in the earthquake forecasting context. Firstly, your suggestion (and that of another reviewer) has prompted us to include a new computational cost section of the appendix. This section reports the training times for each of the models as well as providing details on the computational complexity for training as well as simulating from each model. Encouragingly AutoSTPP and Deep-STPP are quicker to train than ETAS.
>
> There are two reasons which motivate applying ML to earthquake forecasting:
> 1. The current state of real-time forecasting is still quite poor. This is highlighted in the "Pass Rate" of the CSEP tests and commented on in line 485 of the original text.
> 2. Emerging methodologies in data collection have grown the size of earthquake datasets drastically. This provides an ideal oppurtunity for machine learning which performs best in the "big-data" setting. We mention this in line 54 of the original text "Moreover, employing modern techniques, some datasets include smaller magnitude earthquakes, exploring the potential of numerous small events to enhance forecasting performance through flexible NPPs."
>
> The benefits of the ETAS model are
> 1. Its interpretability:
> Its formulation as a branching process provides the ability to distinguish background events from triggered events. We tried to highlight this in line 193 of the original manuscript "estimates the causal structure". To be more explicit about the meaning of this, the updated manuscript replaces this with "distinguishes background events from triggered events".
> Another benefit is that a few of ETAS's nine parameters have been linked to physical properties of the earthquake rupture process. We highlight this in the updated manuscript with the statement: "Several of these learnable parameters have been linked to physical properties of the earthquake rupture process (Utsu et al. 1995; Ide 2013)"
> 2. Its construction using empirical marginal statistical distributions and scalings. We have added the line "This triggering kernel, along with the concept of a background rate, are derived from statistical distributions found through decades of observational studies on short-term earthquake occurence (Utsu 1955; Utsu 1970; Utsu et al., 1995).", to the end of Section 2.2

---

> ### Author Response · Authors · 2024-11-19
> **Response to Reviewer nWWY (3/3)**
>
> > Is there a technical reason for applying only NPPs among the various machine learning methodologies? It would be helpful if you could justify the choice of NPPs by providing a direct technical comparison with other machine learning methods. Have there been any previous studies, or do you have any experimental results, that applied other machine learning methods (e.g., spatio-temporal methods) besides NPPs?
>
> We chose to focus the work on NPPs since point processes are the most predominant model used by the seismology community, in both operational earthquake forecasting (Mizrahi et al., 2024) as well as in existing benchmarking experiments within the seismological community (CSEP) (Taroni et al., 2018; Rhoades et al., 2018) . They are preferred over time-series models (the alternative model class for forecasting) since:
> 1. Time-series require discretisation of time or space (a task which requires finding optimal binning strategies).
> 2. The representation of earthquakes as point process data is consistent with the observation of earthquakes as discrete points in time (Kagan 1994).
> 3. For real-time forecasting, time-series models require the user to wait until the end of the time-bin (hour/day) before the model can be updated, whilst potentially damaging earthquakes can occur. A new statement at the end of Section 2.1 summarises these points to justify our focus on point processes rather than time series models.
>
> However importantly, our inclusion of the CSEP evaluation metrics does in fact allow for time-series models to be benchmarked against our results (by simulating future earthquake occurrences). Your inquiery (and that of another reviewer) has prompted us to clarify that our benchmark is capable of evaluating other classes of model. We now include the following statement in the updated manuscript:
>
> 'This directs the impact of future NPPs to stakeholders in seismology as well broadening the scope of models beyond NPPs: e.g. times series models (Wang et al., 2017) Bayesian approaches (Serafini et al., 2023)'.
>
>
> ### References
>
> Stockman, S., Lawson, D. J., & Werner, M. J. (2023). Forecasting the 2016–2017 Central Apennines earthquake sequence with a neural point process. Earth's Future, 11(9), e2023EF003777.
>
> Utsu, T., (1955). A relation between the area of aftershock region and the energy of mainshock. Journal of the Seismological Society of Japan, 7, 233–240.
>
> Utsu, T. (1970). Aftershocks and earthquake statistics (1): Some parameters which characterize an aftershock sequence and their interrelations. Journal of the Faculty of Science, Hokkaido University, Series 7 (Geophysics), 3(3), 129–195.
>
> Taroni, M., Marzocchi, W., Schorlemmer, D., Werner, M. J., Wiemer, S., Zechar, J. D., ... & Euchner, F. (2018). Prospective CSEP evaluation of 1‐day, 3‐month, and 5‐yr earthquake forecasts for Italy. Seismological Research Letters, 89(4), 1251-1261.
>
> Mizrahi, L., Dallo, I., van der Elst, N. J., Christophersen, A., Spassiani, I., Werner, M. J., ... & Wiemer, S. (2024). Developing, testing, and communicating earthquake forecasts: Current practices and future directions. Reviews of Geophysics, 62(3), e2023RG000823.
>
> Ide, S. (2013). The proportionality between relative plate velocity and seismicity in subduction zones. Nature Geoscience, 6(9), 780-784.
>
> Utsu, T., Ogata, Y., & et al. (1995). The centenary of the Omori formula for a decay law of aftershock activity. Journal of Physics of the Earth, 43(1), 1–33.
>
> Rhoades, D. A., Christophersen, A., Gerstenberger, M. C., Liukis, M., Silva, F., Marzocchi, W., ... & Jordan, T. H. (2018). Highlights from the first ten years of the New Zealand earthquake forecast testing center. Seismological Research Letters, 89(4), 1229-1237.
>
> Kagan, Y. Y. (1994). Observational evidence for earthquakes as a nonlinear dynamic process. Physica D: Nonlinear Phenomena, 77(1-3), 160-192.
>
> Wang, Qianlong, et al. (2017). "Earthquake prediction based on spatio-temporal data mining: an LSTM network approach." IEEE Transactions on Emerging Topics in Computing 8.1
>
> Serafini, F., Lindgren, F., & Naylor, M. (2023). Approximation of Bayesian Hawkes process with inlabru. Environmetrics, 34(5), e2798.

---

> ### Comment · Reviewer_nWWY · 2024-11-27
>
> Thank you for addressing my questions and incorporating the feedback. I appreciate the effort in clarifying your motivations and adding further explanations. These revisions reflect thoughtful engagement with reviewers' comments.
>
> While the research highlights the importance of NPPs in seismology and effectively demonstrates the value of curated datasets, its connection to the machine learning community remains insufficiently explored. The experiments lack diversity in applying machine learning techniques, which limits the paper's relevance to broader researchers in this field.
>
> I acknowledge the significance of this work within seismology, but its relevance to the machine learning community is less clear. A score of 6 remains appropriate, but I am lowering my confidence and leaving the final decision to the AC.

---

> > ### Author Response · Authors · 2024-11-29
> > **Comment on the work's significance to the machine learning community**
> >
> > We are sympathetic to your assessment that our "experiments lack diversity in applying machine learning techniques". This is a direct result on our insistence to use metrics that are most appropriate for earthquake forecasting and therefore limits the scope of directly applicable models. This critical choice of appropriate metrics is discussed in both Section 1.2 and Section 6, and we believe it contributes to the discussion on how to evaluate spatio-temporal models more generally beyond earthquake prediction. Notably, even within the NPP literature, there is no consensus on evaluation metrics beyond log-likelihood, a need explicitly highlighted in Section 7.2 of the review paper by Shchur et al. (2021) .
> >
> > However, we do not agree with the conclusion that the paper's "relevance to the machine learning community is less clear".
> >
> > In contrast, our presentation of our earthquake datasets, along with the associated challenges and best practices in earthquake forecasting, bridges a significant gap that exists between machine learning development and real world application. This gap motivated the creation of the NeurIPS Datasets and Benchmark track (Vanschoren et al., 2021):
> >
> > ".... The vast majority of the NeurIPS community focuses on algorithm design, but often can’t easily find good datasets to evaluate their algorithms in a way that is maximally useful for the community and/or practitioners. Hence, many researchers resort to data that are conveniently available, but not representative of real applications. For instance, many algorithms are only evaluated on toy problems, or data that is plagued with bias, which could lead to biased models or misleading results, and subsequent public criticism of the field (Paullada et al. 2020)."
> >
> > In this context, we argue that our work is relevant to machine learning researchers precisely because it demonstrates how to impact a real world application - something which is not currently achieved by existing benchmarking on the Chen et al., (2021) dataset. Our critique of the Chen et al. (2021) dataset, as discussed in Section 1.1, also contributes to the discourse on data challenges more broadly in spatio-temporal forecasting - critically including how to appropriately partition spatio-temporal data for training, validation and testing.
> >
> > While benchmarking a wider range of machine learning techniques could provide additional insights, such an approach would rely on metrics that are not relevant to seismology, thereby detracting from our primary objective: establishing a practical connection between machine learning and earthquake forecasting. Our focus ensures that the work remains grounded in the real-world requirements of seismology, offering a clear pathway for impactful applications of machine learning.
> >
> >
> > Paullada, A., Raji, I. D., Bender, E. M., Denton, E., & Hanna, A. (2021). Data and its (dis) contents: A survey of dataset development and use in machine learning research. Patterns, 2(11).
> >
> > Shchur, O., Türkmen, A. C., Januschowski, T., & Günnemann, S. (2021). Neural temporal point processes: A review. arXiv preprint arXiv:2104.03528.
> >
> > Vanschoren, J., & Yeung, S. Announcing the NeurIPS 2021 datasets and benchmarks track. 2021. URL https://neuripsconf.medium.com/announcing-theneurips-2021-datasets-and-benchmarks-track-644e27c1e66c.

---

### Official Review · Reviewer_7xrX · 2024-11-03

**Soundness:** 1
**Presentation:** 1
**Contribution:** 2
**Rating:** 3
**Confidence:** 2

**Summary:**

The paper introduces EarthquakeNPP, the benchmark datasets designed to evaluate neural point processes (NPPs) for earthquake forecasting. It compiles earthquake data from California spanning 1971 to 2021 for model assessment. The study highlights that current NPP methods are not yet practical for real-world applications, as none outperform the Epidemic Type Aftershock Sequence (ETAS) model in terms of spatial or temporal log-likelihood performance.

**Strengths:**

1. The work highlights the gap between the seismology and machine learning communities in earthquake prediction, providing potential directions for future development of neural point processes (NPPs) to improve earthquake forecasting.
2. It gathers earthquake data from California from 1971 to 2021, creating a valuable resource for research in both seismology and machine learning.
3. The benchmark datasets and experiments are publicly available, facilitating further research in the field.

**Weaknesses:**

1. The scope of the work is limited. While the paper focuses on NPPs, various deep learning methods [1, 2] can also be applied to earthquake prediction. Including these methods would enhance the comprehensiveness of the evaluation.

2. The paper employs log-likelihood as the evaluation metric. Including metrics such as MAE and accuracy would offer a more comprehensive assessment from multiple perspectives.

3. The description of the benchmark dataset's statistics and characteristics is insufficient. A more detailed analysis, including aspects such as spatial nodes, the evolution of seismic activity, and the pattern similarity between training and testing earthquakes for each dataset, would strengthen the paper.

4. The analysis of model performance is limited. For example, why does Deep-STPP generally perform best in terms of temporal log-likelihood, while AutoSTPP excels in spatial log-likelihood?

5. Terms such as SCEDC_20, SCEDC_25, and SCEDC_30 are unclear and need clarification.

6. The paper's structure is somewhat inconsistent, with descriptions of additional datasets provided without corresponding results in the main text.

[1] Wang, Qianlong, et al. "Earthquake prediction based on spatio-temporal data mining: an LSTM network approach." IEEE Transactions on Emerging Topics in Computing 8.1 (2017): 148-158.
[2] Vardaan, K., et al. "Earthquake trend prediction using long short-term memory RNN." International Journal of Electrical and Computer Engineering 9.2 (2019): 1304-1312.

**Questions:**

1. Why does the paper focus only on neural point processes and exclude other model types?
2. Could you justify the use of log-likelihood as the primary metric while excluding other metrics such as MAE and R-score?
3. Could you provide a more detailed description of the benchmark dataset's statistics and characteristics?
4. Could you clarify the meaning of dataset labels such as SCEDC_20, SCEDC_25, and SCEDC_30? Do these refer to SCEDC data with magnitude thresholds of 2.0, 2.5, and 3.0, respectively?
4. Is there experimental evidence indicating that the degraded performance of NPPs compared to ETAS is due to their lack of direct dependence on magnitudes, rather than other factors?

---

> ### Author Response · Authors · 2024-11-19
> **Response to Reviewer 7xrX (1/3)**
>
> We thank the reviewer for taking the time to read our paper and for such valuable feedback. Below we provide our responses in a point-by-point manner:
>
> > .... various deep learning methods [1, 2] can also be applied to earthquake prediction. Including these methods would enhance the comprehensiveness of the evaluation. Why does the paper focus only on neural point processes and exclude other model types?
>
> Thank you for raising these time-series papers, they are important to discuss because _these papers are not properly benchmarked_. Specifically they were not benchmarked against state-of-the-art ETAS (or in fact against any benchmark model!) and so there is no way to know if they are competitive.
>
> We chose to focus the work on NPPs, since point processes are the most predominant model used by the seismology community, in both real-time operational earthquake forecasting by (non) government agencies (Mizrahi et al., 2024) as well as in existing benchmarking experiments within the seismological community (CSEP) (Taroni et al., 2024; Rhoades et al., 2018) . They are preferred over time-series models since:
> 1. Time-series require discretisation of time or space (a task which requires finding optimal binning strategies).
> 2. The representation of earthquakes as point process data is consistent with the observation of earthquakes as discrete points in time (Kagan 1994). For example, an earthquake changes the stress field near-instantaneously (at the speed of sound in the Earth's crust) and therefore point process approaches are ideally suited to capture the immediate change in conditions leading to aftershocks and triggering of earthquakes.
> 3. For real-time forecasting, time-series models require the user to wait until the end of the time-bin (hour/day) before the model can be updated, whilst potentially damaging earthquakes can occur. A new statement at the end of Section 2.1 summarises these points to justify our focus on point processes rather than time series models.
>
> However, our inclusion of the CSEP evaluation metrics allows for time-series models to be benchmarked against our results (by simulating future earthquake occurrences) and we encourage the authors of the papers you reference to use our platform. Your enquiry (and that of another reviewer) has prompted us to clarify that our benchmark is capable of evaluating other classes of model. We now include the following statement in the updated manuscript:
>
> 'This directs the impact of future NPPs to stakeholders in seismology as well broadening the scope of models beyond NPPs: e.g. times series models (Wang et al., 2017), Bayesian approaches (Serafini et al., 2023) '
>
> > Could you justify the use of log-likelihood as the primary metric while excluding other metrics such as MAE and R-score?
>
> Whilst the log-likelihood is the primary evaluation metric, the additional CSEP evaluation metrics enable a more comprehensive assessment and enable benchmarking of broader classes of models (generative NPPs, time-series, Bayesian methods).
>
> We discuss our omission of the MAE metric in detail in both Section 1.2 and Section 6 of the original manuscript.
>
> "Root Mean Square Error (RMSE) and related scores (MAE) are considered to be flawed and misleading for seismological prediction (Hodson, 2022), because the predictive distribution is strongly skewed and therefore far from the Gaussian or Laplacian errors that RMSE and MAE are designed for."
>
> " Probabilistic seismic hazard analysis (PSHA) requires long-term prediction beyond the next-event (Ebrahimian et al., 2014; Gerstenberger et al., 2014), therefore this approach (CSEP evaluation) offers stakeholders a more comprehensive understanding of earthquake hazard than metrics focused on predicting the next event (e.g. RMSE, MAE). The procedure also follows the recommendation by Shchur et al. (2021) to move away from next-event point prediction (MAE) for NPPs."
>
> The "R-score" you refer to is not a valid metric for benchmarking since it is a measure of correlation. It is used by [2] to measure correlation between input features and the output predictor in time-series forecasting. Not only would including this metric require all benchmarked models to be time-series, but to use it as an evaluation metric would require all time-series models to have identical input features - this is not even the case for the two time-series models you have cited.

---

> ### Author Response · Authors · 2024-11-19
> **Response to Reviewer 7xrX (2/3)**
>
> > The description of the benchmark dataset's statistics and characteristics is insufficient. A more detailed analysis, including aspects such as spatial nodes, the evolution of seismic activity, and the pattern similarity between training and testing earthquakes for each dataset, would strengthen the paper.
>
> Beyond the dataset descriptions found in Section 3, further dataset information can be found within the [Datasets](https://anonymous.4open.science/r/EarthquakeNPP-2D51/Datasets/README.md) directory of the anon. github repository. The [root](https://anonymous.4open.science/r/EarthquakeNPP-2D51/Datasets/README.md) contains details on the dataset format including field descriptions as well as providing a table detailing how each dataset is partitioned for training/validation/testing, the size of the training/testing set and the magnitude thresholds. Within the [subdirectory](https://anonymous.4open.science/r/EarthquakeNPP-2D51/Datasets/ComCat/README.md) of each corresponding dataset is a jupyter notebook detailing how each dataset is accessed and preprocessed for our benchmark. Within each notebook multiple plots are produced for each dataset, including the spatial locations of all earthquakes and the evolution of the dataset over time (where the training and testing sections are colour coded).
>
> Your (and other reviewers) comments have prompted us to make this information more accessible by copying it to the text in an updated manuscript. To enhance our analysis of the datasets, we have added further plots of each dataset: a plot of the cumulative number of events evolving over time as well as colouring the locations of the events by their dataset partition (training/validation/testing).
>
> Please could you clarify what you mean by "spatial nodes"? Since the models are point processes, there is no need to discretise the spatial domain for input into the models. If you are referring to the spatial discretisation used for the Spatial Test (Section 5.2), then the gridding used can be found in the anon. github repository [here](https://anonymous.4open.science/r/EarthquakeNPP-2D51/Experiments/ETAS/README.md). However if "spatial nodes" refers to seismic stations, we do not believe that plotting them aids with understanding the datasets in any way.
>
> > The analysis of model performance is limited. For example, why does Deep-STPP generally perform best in terms of temporal log-likelihood, while AutoSTPP excels in spatial log-likelihood?
>
> That's a good suggestion; we have added the following analysis/interpretation in Section 4.
>
> We speculate that AutoSTPP performs best in spatial log-likelihood, since its model formulation can capture anisotropic Hawkes kernels (Figure 2 of Zhou et al., 2024) which are often observed in earthquake data (Page et al., 2022). We speculate that DeepSTPP performs best in terms of temporal log-likelihood since its formulation accounts for the influence from unobserved events - a phenomenon that varies temporally in earthquake data (See Section A.2 Earthquake Catalog Completeness from original manuscript).
>
> In this work, we chose to concentrate our efforts on the presentation of datasets and seismology concepts to a machine learning audience, leaving little space for the further analysis that should be done interpreting model performance (this is often the subject of a whole paper in seismology e.g. Stockman et al., 2023). However, your suggestion to include more analysis has prompted us to provide a forecasting "case study" that can be used as an example for how to quantitatively & qualitatively interpret forecasting results in a more fine grain manner.
>
> A new section of the appendix (titled "2019 M7.2 Ridgecrest Earthquake Case Study") demonstrates how the CSEP metrics can be used to analyse a model's performance during the 2019 Ridgecrest earthquake sequence (the most recent challenging real-time forecasting scenario for Californian agencies). Whilst this demonstration is only for the ETAS model, it will encourage future ML works to perform such qualitative analysis using our benchmark datasets.
>
> > Terms such as SCEDC_20, SCEDC_25, and SCEDC_30 are unclear and need clarification....
>
> Apologies, we should have defined these terms in the original manuscript. You are correct, _SCEDC_20_, _SCEDC_25_, and _SCEDC_30_ do refer to the three magnitude thresholds of the SCEDC catalog. The updated manuscript defines these terms in the caption of Figure 3.

---

> ### Author Response · Authors · 2024-11-19
> **Response to Reviewer 7xrX (3/3)**
>
> > The paper's structure is somewhat inconsistent, with descriptions of additional datasets provided without corresponding results in the main text.
>
> We agree that the description of the additional datasets would be better placed in the appendix with their corresponding results. The updated manuscript now makes this restructuring.
>
> > Is there experimental evidence indicating that the degraded performance of NPPs compared to ETAS is due to their lack of direct dependence on magnitudes, rather than other factors?
>
> Producing this experimental evidence would require implementing magnitude dependence for all the NPPs in our benchmark. This would require modeling choices to be made for every single model, including a discussion and description of these choices. Whilst this is absolutely a "next-step" for this work, we believe this is well beyond the scope and distracts from our presentation of datasets and seismological concepts to a machine learning audience.
>
> Whilst we don't have any direct experimental evidence for this speculation, it is well accepted in seismology that the magnitudes of past earthquakes significantly affects the rate of future earthquakes. Seminal empirical studies (Utsu & Seki, 1955; Utsu, 1970) show this, and as a result, all competitive short-term earthquake forecasting models use this magnitude information (Taroni et al., 2018; Mizrahi et al., 2024).
>
> It was our hope that the references we cited in line 516 of the original manuscript provided enough evidence that magnitude information is a significant predictor of future earthquake rates. To emphasise this point further, we have added a further statement:
>
> "A consequence of these important observational studies is that all competitive short-term earthquake forecasting models used operationally (Mizrahi et al., 2024) or tested by CSEP (Taroni et al., 2018) use magnitude information for their forecasts."
>
> ### References
>
> Mizrahi, L., Dallo, I., van der Elst, N. J., Christophersen, A., Spassiani, I., Werner, M. J., ... & Wiemer, S. (2024). Developing, testing, and communicating earthquake forecasts: Current practices and future directions. Reviews of Geophysics, 62(3), e2023RG000823.
>
> Taroni, M., Marzocchi, W., Schorlemmer, D., Werner, M. J., Wiemer, S., Zechar, J. D., ... & Euchner, F. (2018). Prospective CSEP evaluation of 1‐day, 3‐month, and 5‐yr earthquake forecasts for Italy. Seismological Research Letters, 89(4), 1251-1261.
>
> Rhoades, D. A., Christophersen, A., Gerstenberger, M. C., Liukis, M., Silva, F., Marzocchi, W., ... & Jordan, T. H. (2018). Highlights from the first ten years of the New Zealand earthquake forecast testing center. Seismological Research Letters, 89(4), 1229-1237.
>
> Zhou, Z., & Yu, R. (2024). Automatic integration for spatiotemporal neural point processes. Advances in Neural Information Processing Systems, 36.
>
> Page, M. T., & van der Elst, N. J. (2022). Aftershocks preferentially occur in previously active areas. The Seismic Record, 2(2), 100-106.
>
> Kagan, Y. Y. (1994). Observational evidence for earthquakes as a nonlinear dynamic process. Physica D: Nonlinear Phenomena, 77(1-3), 160-192.
>
> Wang, Qianlong, et al. (2017). "Earthquake prediction based on spatio-temporal data mining: an LSTM network approach." IEEE Transactions on Emerging Topics in Computing 8.1
>
> Serafini, F., Lindgren, F., & Naylor, M. (2023). Approximation of Bayesian Hawkes process with inlabru. Environmetrics, 34(5), e2798.
>
> Stockman, S., Lawson, D. J., & Werner, M. J. (2023). Forecasting the 2016–2017 Central Apennines earthquake sequence with a neural point process. Earth's Future, 11(9), e2023EF003777.
>
> Utsu, T., & Seki, A. (1955). A relation between the area of aftershock region and the energy of mainshock. Journal of the Seismological Society of Japan, 7, 233–240.
>
> Utsu, T. (1970). Aftershocks and earthquake statistics (1): Some parameters which characterize an aftershock sequence and their interrelations. Journal of the Faculty of Science, Hokkaido University, Series 7 (Geophysics), 3(3), 129–195.

---

> > ### Comment · Reviewer_7xrX · 2024-11-26
> >
> > Thank you for the authors' response. I am not an expert in seismology, but I understand this work aims to bridge the gap between the seismology and machine learning communities in earthquake prediction. To me, the lack of comparison with related works from the machine learning community and the expectation that they use the proposed platform for comparison themselves is not convincing. While I agree that RMSE is unsuitable due to the strongly skewed predictive distribution, this is not inherently true for MAE. Moreover, the referenced paper (Hodson, 2022) critiques RMSE but does not argue that MAE is unsuitable.
> >
> > However, given my limited expertise in seismology, I will lower my confidence and would like to leave this to be decided by ACs.

---

> > > ### Author Response · Authors · 2024-11-26
> > > **Omission of MAE Metric**
> > >
> > > Thank you for engaging in discussion on this subject and revealing important details that need addressing. You have highlighted that we have still failed to provide sufficient justification for our omission of the MAE metric (and therefore the additional machine learning models). An updated manuscript (now uploaded) expands on our original point.
> > >
> > > As Hodson et al., (2022) outline, RMSE is suitable for normally distributed errors (found when prediction and observation are normally distributed), whereas MAE is suitable for Laplacian errors (found when prediction and observation are exponentially distributed). Whilst an exponential is indeed skewed, observational studies on earthquakes have shown that both earthquake times (Kagan 1994) and earthquake locations (Felzer et al., 2006) obey power law distributions. These distributions are heavy-tailed and _highly_ skewed and therefore errors are not Laplacian.
> > >
> > > To demonstrate this, we have added a new Figure 22 to the the appendix. The figure plots the distribution of the errors from Normal, Exponential and Pareto (a heavy-tailed, power-law distribution). Whilst Normal and Laplacian distributions (fit by maximum likelihood) can approximate the errors from Normal and Exponential distributions, they cannot approximate the errors from the Pareto.
> > >
> > > The end of Section 1.2 has been updated with this additional detail and references the new Figure 22:
> > >
> > > "Traditional metrics like Root Mean Square Error (RMSE) and Mean Absolute Error (MAE) are inadequate and potentially misleading for seismological predictions (Hodson, 2022), as earthquake occurrence follows power law distributions (Kagan, 1994; Felzer et al., 2006) that are heavy-tailed, making the errors non-Gaussian and non-Laplacian — contrary to the assumptions underlying RMSE and MAE (see Section G)."
> > >
> > > Kagan, Y. Y. (1994). Observational evidence for earthquakes as a nonlinear dynamic process. Physica D: Nonlinear Phenomena, 77(1-3), 160-192.
> > > Felzer, K. R., & Brodsky, E. E. (2006). Decay of aftershock density with distance indicates triggering by dynamic stress. Nature, 441(7094), 735-738.

---

### Official Review · Reviewer_PNTo · 2024-11-03

**Soundness:** 3
**Presentation:** 2
**Contribution:** 2
**Rating:** 6
**Confidence:** 2

**Summary:**

This work introduces EarthquakeNPP, a benchmark dataset for earthquake forecasting using Neural Point Processes (NPPs), aiming to enhance existing models. It covers seismic records from 1971 to 2021 in various California regions, including timestamps, coordinates, and magnitudes. Comparing NPPs with the ETAS model, it finds that current NPP implementations do not outperform ETAS in spatial or temporal log-likelihood, indicating that NPPs are not yet ready for practical earthquake forecasting. However, EarthquakeNPP serves as a platform for collaboration between the seismology and machine learning communities to improve predictability.

**Strengths:**

(1) The dataset spans multiple regions in California over an extended period and includes earthquakes of various magnitudes, including smaller ones. This enhances data diversity and increases the dataset's granularity.

(2) It provides a direct comparison with the popular ETAS model, promoting collaboration between the seismology and machine learning communities.

(3) By utilizing modern seismic detection technologies, the dataset offers improved data quality.

**Weaknesses:**

(1) The application of NPPs in practical earthquake forecasting is limited. For example, the high computational cost of generating repeated sequences may hinder real-world applications. I noticed that you used four A100 GPUs for training. Have you conducted any computational complexity analysis, such as comparing the training times between the baseline and NPPs? If you have other ways to present the computational cost, please include them as well.

(2) This work lacks sufficient detail on the reproducibility of experimental results, which may affect other researchers' ability to replicate the study. For instance, issues with the division of training and testing data in the existing NPPs benchmark dataset may prevent an accurate reproduction of real operational environments. Additionally, the interpretation of experimental results relies heavily on statistical data, lacking an in-depth analysis of the underlying physical mechanisms.

**Questions:**

Refer to weaknesses

---

> ### Author Response · Authors · 2024-11-19
> **Response to Reviewer PNTo (1/2)**
>
> We thank the reviewer for taking the time to read our paper and for such valuable feedback. Below we provide our responses in a point-by-point manner:
>
> > ... Have you conducted any computational complexity analysis, such as comparing the training times between the baseline and NPPs? If you have other ways to present the computational cost, please include them as well.
>
> We agree that computational cost/complexity is essential information to determine real-world applicability (as well as further enhancing reproducibility as per your next point). Following your suggestion, we have now included training times and computational complexity in the appendix. Encouragingly, both AutoSTPP and Deep-STPP are quicker to train than ETAS since they benefit from GPU acceleration (unlike ETAS) and use a sliding window of the recent history of events (unlike ETAS which uses the entire history of events). NSTPP on the other hand is considerably slower than all other models, since training involves solving an ODE.
>
> A new 'Computational Cost' section of the appendix provides this additional information for each model and presents some discussion on the real-world applicability as a consequence.
>
> > .....high computational cost of generating repeated sequences may hinder real-world applications
>
> Unfortunately the procedure of generating repeated sequences has high computational cost even for simple parametric models such as ETAS, since at least 10,000 sequences must be generated. The new 'Computational Cost' section of the appendix provides a discussion on the cost of simulation for each model.
>
>
> > This work lacks sufficient detail on the reproducibility of experimental results... For instance, issues with the division of training and testing data in the existing NPPs benchmark dataset may prevent an accurate reproduction of real operational environments.
>
> We tried to enhance the reproducibility of our work through documentation found within the anon. github repository. All results can be reproduced by running the commands found in [README.md](https://anonymous.4open.science/r/EarthquakeNPP-2D51/README.md). For new models wishing to benchmark using the datasets, further information on how the data is divided for training/validation/testing can be found in a table in [Datasets/README.md](https://anonymous.4open.science/r/EarthquakeNPP-2D51/Datasets/README.md), is further illustrated in figures of each dataset ([found here](https://anonymous.4open.science/r/EarthquakeNPP-2D51/Datasets/ComCat/README.md)), and is also provided in the config files of each model. You (and other reviewers) have highlighted that this information would be more easily accessible if it was also found in the manuscript. As such, a table copied to the main text explicitly defines the train/val/test splits and additional content in the appendix plots each dataset and gives information on the storage format, including field descriptions.

---

> ### Author Response · Authors · 2024-11-19
> **Response to Reviewer PNTo (2/2)**
>
> > Additionally, the interpretation of experimental results relies heavily on statistical data, lacking an in-depth analysis of the underlying physical mechanisms.
>
> We completely agree that physical interpretations would provide much more insight into the results of the benchmarking experiment. Given that the models themselves are statistical, we cannot directly query any physical process driving the results. Instead we can only speculate using known physical differences between the datasets:
>
> _QTM_SaltonSea_, _QTM_SanJac_ and _White_ datasets cover the Salton Sea and San Jacinto subregions of California - areas know to have a high presence of earthquakes swarms. Earthquake swarms are transient accelerations in seismicity (Llenos et al., 2019) thought to be driven by external processes, such as fluid flow. Such phenomena are known to be poorly captured by "aftershock" models such as ETAS and we speculate that they could be driving greater performance of NPPs relative to ETAS.
>
> Furthermore, the _ComCat_ dataset includes the northern section of the San Andreas fault which contains the Mendecino Tripple Junction (Hellweg et al., 2024), where the Gorda, North American and the Pacific plate all meet. The three plates moving at different velocities create a very complex fault system with a high level of seismic activity. We speculate that these complex dynamics are not as well captured by typical models such as ETAS, resulting in better relative NPP performance than for the SCEDC catalog for only Southern California.
>
> In this work, we chose to concentrate our efforts on the presentation of datasets and seismological concepts to a machine learning audience. As such, a conference paper leaves little extra room for an in depth analysis of the results. We believe this analysis is important next steps for the adoption of NPPs into earthquake forecasting and would allow for exploration of the physical processes briefly mentioned above. We would prefer to keep the above speculations brief in the main text with the sentence.
>
> "Better relative temporal performance of NPPs for ComCat, QTM_SaltonSea, QTM_SanJac, and White datasets may be driven by more complex physical processes such as earthquake swarms (Llenos et al., 2019) or the Mendecino triple junction (Hellweg et al., 2024)."
>
> ### References
>
> Llenos, A. L., & van der Elst, N. J. (2019). Improving earthquake forecasts during swarms with a duration model. Bulletin of the Seismological Society of America, 109(3), 1148-1155.
>
> Hellweg, M., Dreger, D. S., Lomax, A., McPherson, R. C., & Dengler, L. (2024). The 2021 and 2022 North Coast California Earthquake Sequences and Fault Complexity in the Vicinity of the Mendocino Triple Junction. Bulletin of the Seismological Society of America. doi: https://doi.org/10.1785/0120240023

---

> ### Comment · Reviewer_PNTo · 2024-11-26
>
> Thanks for the rebuttal, I will keep my score because a score of 6 is relatively neutral. Giving an 8 requires a deep understanding of this field, and I'm not confident enough to give an 8.

---

### Official Review · Reviewer_Qezt · 2024-11-04

**Soundness:** 3
**Presentation:** 2
**Contribution:** 2
**Rating:** 6
**Confidence:** 3

**Summary:**

The authors have presented EarthquakeNPP, a group of datasets designed to evaluate/benchmark NPP models for earthquake forecasting. The authors evaluate some state-of-the-art NPP-based models against ETAS and have identified that these models demonstrate comparable temporal performance with the ETAS  model. Based on their analysis, the authors note that NPP-based models do not have a direct dependence on magnitude, in contrast to ETAS.

**Strengths:**

**Originality**: While all the datasets are already available, the authors have created a pipeline and baseline and state-of-the-art methods and evaluation metrics to formalize the process of evaluating earthquake forecasting methods. Standardizing the evaluation method by comparing against the ETAS method that is used in the real world is a good idea.

**Significance**: The dataset collection created by the authors can serve as a benchmark for future research in NPP-based earthquake forecasting models. The authors have highlighted the limitations of existing benchmark datasets and why they need to be improved.

**Experiments**: The authors have also provided initial assessments of state-of-the-art NPP-based models and highlighted the limitations of existing methods.

**Weaknesses:**

Clarity:
1. Section 1.1 The three categories defined in related work are hard to follow. A table highlighting the differences/limitations, or better defining the categories would help. For instance, "benchmarking efforts by NPP/ML community" , "benchmarking efforts by seismology community" etc. The existing benchmark dataset can then go into the first category.
2. Section 5: Since each of the three tests is aimed at evaluating the temporal, spatial, and magnitude components of the forecast, renaming section 5.1 to "temporal test" is suggested.
3. Section 3 is better summarized as a table and includes additional information such as the number of training, validation, and test data. (This is currently included in the anon. github README). Additionally, the strengths/weaknesses/challenges of each dataset can be elucidated in this format, allowing the user to understand how each dataset is different from the other.
4. From the title, it seems as if the dataset is purely for benchmarking NPP-based methods, which might narrow the scope of the dataset. What needs to be done to generalize it for other methods capable of generating log-likelihoods such as bayesian inferencing-based methods?

**Questions:**

1. Is it possible that since ETAS was trained using both training + validation data while the NPP models were trained on the training data, it is performing better? What is the impact of training data size on the evaluated models? Please conduct experiments such as evaluating ETAS using only training data.
2. Including a comparison of model complexity of the NPP-based models vs ETAS models in the evaluation metrics would be of great importance.

---

> ### Author Response · Authors · 2024-11-19
> **Response to Reviewer Qezt (1/2)**
>
> We thank the reviewer for taking the time to read our paper and for such valuable suggestions. Below we provide our responses in a point-by-point manner:
>
> > ... A table highlighting the differences/limitations, or better defining the categories would help. For instance, "benchmarking efforts by NPP/ML community" , "benchmarking efforts by seismology community" .....
>
> Thank you for suggestions, we agree that the structure of Section 1.1 made it hard to follow and that redefining the categories to "benchmarking efforts by NPP/ML community" and "benchmarking efforts by seismology community" would improve clarity. Following your other suggestion (mirrored by another reviewer), we shall also include a small table summarising the differences/improvements of our datasets compared with the previous.
>
> | Dataset       | Chronological Training/Test Splits | Complete Timespan | Complete Magnitudes | Used by Local Agencies |
> |---------------|-----------------------|----------------------------|----------------------|-------------------------|
> | Chen et al. (2021) Dataset   | ❌                   | ❌                         | ❌                   | ❌
> | EarthquakeNPP Datasets  | ✔️                   | ✔️                         | ✔️                   | ✔️       |
>
>
> > Renaming section 5.1 to "temporal test" is suggested.
>
> We agree that "temporal test" would be a clearer name for the test as it is presented in this work. The name "Number test" comes from the test counting the _number_ of earthquakes in a discrete time interval (days) to evaluate the temporal component of the forecast. Since the name of the test was established in 2007 as part of CSEP's activities, we cannot change its name. However, to clarify its nature as temporal evaluation, we shall rename section 5.1 "Number (Temporal) Test"
>
> > Section 3 is better summarized as a table and includes additional information such as the number of training, validation, and test data....
>
> We completely agree that Section 3 could be more efficiently summarised as a table, particularly moving the table you reference from the anon. github README to the main text. This will be included in the updated manuscript, with an additional "Description" column that outlines the important characteristics of each dataset. Your suggestions, along with those of other reviewers has also prompted us to move further dataset information from the github repo into the appendix. This includes some plots of each dataset, including key earthquakes, as well as the format of the dataset with field descriptions.
>
> > From the title, it seems as if the dataset is purely for benchmarking NPP-based methods, which might narrow the scope of the dataset...
>
> Our initial motivation to create the dataset was to benchmark recently developed NPPs alongside models from the seismology community, particularly due to recent excitement around NPPs and the presence of a poorly constructed existing benchmark. Our benchmarking experiment is conducted using the log-likelihood evaluation metric and therefore any other point process model (with a well defined likelihood) can be benchmarked alongside our results. To broaden the scope of our datasets and to enable the evaluation of a much larger range of model classes, we further added the CSEP evaluation tests. These tests do not require models to posses a likelihood and instead only requires them to simulate future earthquakes. This allows a huge variety of forecasting models to be benchmarked against our results for the ETAS model. This would include time-series models (Wang et al., 2017), bayesian models (Serafini et al., 2023) and generative point process models (Yuan et al., 2023; Li et al., 2024).
>
> We agree that the title NPP suggests a smaller scope of models than the datasets are capable of benchmarking. However, since our intention is to target the NPP community and to replace the poor existing benchmark, we feel this title is most impactful. To make it more explicit that our benchmark is capable of further model classes, we include an additional statement at the beginning of the Introduction:
>
> 'This directs the impact of future NPPs to stakeholders in seismology as well broadening the scope of models beyond NPPs: e.g. times series models (Wang et al., 2017) Bayesian approaches (Serafini et al., 2023)'

---

> ### Author Response · Authors · 2024-11-19
> **Response to Reviewer Qezt (2/2)**
>
> > Is it possible that since ETAS was trained using both training + validation data while the NPP models were trained on the training data, it is performing better?....
>
> This is a good suggestion for understanding the improved performance of ETAS over the NPPs. As a result, we retrained ETAS on only training data and found no significant difference in its performance relative to the NPPs. Whilst this is useful to investigate the cause of performance difference, we believe that for valid benchmarking, all models should be "exposed" to the same data during training. Since ETAS is a parametric model with only 9 parameters, it is not susceptible to overfitting and therefore doesn't require validation data to control for this. This is unlike NPPs, which would overfit if purely trained on combined training + validation data.
>
> > Including a comparison of model complexity of the NPP-based models vs ETAS models in the evaluation metrics would be of great importance.
>
> We agree that model complexity/training time is essential for evaluating the utility of these models, especially if they are to be deployed operationally. As per your suggestion, the appendix now includes a table recording the training times and complexity of each model with a brief discussion on the results, e.g. NSTPP is far too computationally expensive to be used in practice, Deep-STPP and AutoSTPP are faster to train than ETAS since they benefit from GPU acceleration (unlike ETAS) and use a sliding window of the recent history of events (unlike ETAS which uses the entire history of events). We also believe that your suggested inclusion improves the reproducibility of our benchmark, since users can anticipate the training time of each model if they wish to recreate our results.
>
> ### References
>
> Wang, Qianlong, et al. (2017). "Earthquake prediction based on spatio-temporal data mining: an LSTM network approach." IEEE Transactions on Emerging Topics in Computing 8.1
>
> Serafini, F., Lindgren, F., & Naylor, M. (2023). Approximation of Bayesian Hawkes process with inlabru. Environmetrics, 34(5), e2798.
>
> Yuan Yuan, Jingtao Ding, Chenyang Shao, Depeng Jin, and Yong Li. (2023). Spatio-temporal diffusion point processes. In Proceedings of the 29th ACM SIGKDD Conference on Knowledge Discovery and Data Mining, pp. 3173–3184.
>
> Li, Z., Xu, Q., Xu, Z., Mei, Y., Zhao, T. &amp; Zha, H.. (2024). Beyond Point Prediction: Score Matching-based Pseudolikelihood Estimation of Neural Marked Spatio-Temporal Point Process. Proceedings of the 41st International Conference on Machine Learning, in Proceedings of Machine Learning Research 235:29096-29111 Available from https://proceedings.mlr.press/v235/li24cb.html.

---

> > ### Comment · Reviewer_Qezt · 2024-11-26
> > **Re: model complexity**
> >
> > Thank you for answering my questions. Regarding model complexity, including the number of trainable parameters & model size for each method is a better way to show model complexity in lieu of training time.

---

> > > ### Author Response · Authors · 2024-11-26
> > > **Re: model complexity**
> > >
> > > Thank you for your additional suggestions regarding model complexity. These will be included in an updated version of the manuscript.

---

### Author Response · Authors · 2024-11-22
**Updated manuscript has been uploaded.**

We sincerely thank all reviewers for their thoughtful feedback and valuable suggestions. We have now uploaded an updated version of the manuscript in which we have carefully considered each comment and made significant updates to the paper as a result. We have made both minor changes in response to the comments, as well as the following major changes:

1. Section 3 which introduces our datasets has been instead summarised as a table. (Reviewer: Qezt)
2. An expanded discussion of results has been included at the end of Section 4. (Reviewer: PNTo, 7xrX)
3. A "Computational Efficiency" section has been included as appendix Section C. (Reviewer: Qezt, PNTo, nWWY)
4. A justification on our focus on point processes has been included at the end of Section 2.1. (Reviewer: Qezt, 7xrX, nWWY)
5. The abstract and the end of introduction (Section 1) now clarify that our CSEP metrics broaden the scope of models beyond point processes. (Reviewer: Qezt, 7xrX, nWWY)
6. We have copied dataset information from the anon. github repository to the text. This includes dataset statistics included in the table in Section 3 and a new "Further Dataset Figures" appendix Section E. (Reviewer: Qezt, PNTo, 7xrX)
7. A new appendix, Section F, titled "2019 M7.1 Ridgecrest Earthquake Case Study," presents a real-time forecasting example, illustrating how to apply CSEP metrics for more qualitative analysis. (Reviewers: 7xrX, nWWY)

---

### Meta-Review · Area_Chair_4doF · 2024-12-22

**Metareview:**

The paper presents a new benchmark for earthquake forecasting by collecting datasets of various seismic records in different California regions across decades. The paper then specifically evaluates NPP models and show that these models show good performance compared to a current SOTA statistical model called ETAS, but fall short demonstrating the value of the benchmark in practical utility of NPP models.

Strengths: comprehensive collection of datasets and evaluation pipeline/workflow/metrics that can serve as a useful benchmark for seismology ML researchers

Weaknesses: The main weakness raised was that the paper was either narrow in its scope to focus only on NPP models without enough experimental evidence to discard other models or that the narrow focus on NPP was also limited in deeper analyses of these models.

Points of improvement suggested:
* Broadened focus:  It would be beneficial to include other types of DL models as possible benchmarks (time series models and general sequence models). If these models are subpar by design, then the benchmark created by the authors would be an ideal vehicle to demonstrate this.
* Alternatively, the focus could be narrowed to NPPs but the analysis expanded on their performance. Suggested ideas were:

    - Analysis on which period, time, and location the NPPs models predicted accurately/inaccurately
    - Analysis connecting underlying physical mechanisms to the performance and in general, an analysis of the performance difference
      between NPP models and how they connect to the underlying physics
    - Real-world earthquake forecasting and response scenarios using NPPs (while this is partially in the revised manuscript with the case
      study, it does not evaluate NPPs)

**Additional Comments On Reviewer Discussion:**

The main weakness was that the paper focused on NPPs as the only DL models capable of the earthquake forecasting task, without including other models that the benchmark could have been used to demonstrate their reduced value. Alternatively, it is fair to narrow the focus onto a specific class of models (NPPs here) with an appropriate discussion and perform a more in-depth analysis on the performance of these models on the benchmark (see suggested ideas above). However, the reviewers felt the paper fell a little short on doing either and hence there was no clear champion of the paper. After the rebuttal, two reviewers further lowered their confidence due to the same reason.

Other questions raised by the reviewers included increased clarity in presentation of the dataset, more explanations in choice of metrics, computational complexity details. The reviewers had several suggestions to improve the paper clarity and presentation that I believe the authors have satisfactorily addressed these concerns with an improved revision.

All reviewers agree in the inherent value of the dataset and its usefulness to the community, with 3 score 6s and 1 score 3 (reviewer 7xrX).  7xrX states "To me, the lack of comparison with related works from the machine learning community and the expectation that they use the proposed platform for comparison themselves is not convincing" after the rebuttal without a change in score but reduced confidence due to lack of domain knowledge in seismology.

Incorporating some of the suggestions in (a) more diversity in evaluated models or (b) more analyses on the NPP models with connections to real forecasting case studies would make the paper a strong contribution.

---

### Decision · Program_Chairs · 2025-01-22

Reject